# UPCYCLING INSTRUCTION TUNING FROM DENSE TO MIXTURE-OF-EXPERTS VIA PARAMETER MERGING

## ABSTRACT

Mixture-of-Experts (MoE) shines brightly in large language models (LLMs) and demonstrates outstanding performance in plentiful natural language processing tasks. However, existing methods transforming LLMs from dense to MoE face significant data requirements and typically rely on large-scale post-training. In this paper, we propose Upcycling Instruction Tuning (UpIT), a data-efficient approach for tuning a dense pre-trained model into a MoE instruction model. Specifically, we first point out that intermediate checkpoints during instruction tuning of the dense model are naturally suitable for specialized experts, and then propose an expert expansion stage to flexibly achieve models with flexible numbers of experts, where genetic algorithm and parameter merging are introduced to ensure sufficient diversity of new extended experts. To ensure that each specialized expert in the MoE model works as expected, we select a small amount of seed data that each expert excels to pre-optimize the router. Extensive experiments with various data scales and upcycling settings demonstrate the outstanding performance and data efficiency of UpIT, as well as stable improvement in expert or data scaling. Further analysis reveals the importance of ensuring expert diversity in upcycling.

## 1 INTRODUCTION

Large Language Models (LLMs) have demonstrated remarkable performance on various NLP tasks and are gradually becoming part of our daily lives through chatbot applications such as ChatGPT, Copilot, etc (Ouyang et al., 2022; Touvron et al., 2023; OpenAI, 2024). As LLMs become increasingly prevalent, the high computational of traditional dense architecture with high computational costs in the inference phase poses significant obstacles to downstream deployment. How to improve the model performance without proportionally increasing computing resources become a hot topic in the field (Muennighoff et al., 2024; Xue et al., 2024). In response to this challenge, Mixture-of-Experts (MoE) receives extensive attention due to its excellent scalability, which expands model capacity with almost no extra inference overhead (Fedus et al., 2022; Zoph et al., 2022). Recently, many MoE-based LLMs have emerged in various scenarios with outstanding effectiveness and efficiency (Dai et al., 2024; Jiang et al., 2024; Zhu et al., 2024a).

Upcycling is garnering increasing attention for converting dense LLMs through a series of processes, including expanding experts, integrating routers, and subsequent post-training, ultimately yielding MoE-style models. As depicted in Figure 1, current solutions are broadly classified into two categories: (a) **Vanilla Upcycling**, which directly upcycle a dense model to a MoE model by replicating FFN layers, followed by a large-scale post-training to optimize the additional experts and corresponding routers (Komatsuzaki et al., 2023). Due to the homogeneity of experts in the initial stage, a large amount of post-training data is usually necessary, such as ~1T tokens for full parameter training or ~5M instruction data for parameter efficient fine-tuning (Dou et al., 2024; Zhu et al., 2024a). (b) **Specialized Upcycling**, which first trains specialized experts based on meticulously designed domain-specific datasets and then proceeds with upcycling and post-training (Sukhbaatar et al., 2024). Despite having superior performance, it still requires hundreds of billions of elaborately constructed domain data and lacks flexibility in scaling the number of domain-specific experts. To sum up, although expert specialization slightly reduces data requirements, the current approach to upcycling from dense to MoE heavily relies on a large amount of training data, *how to efficiently and flexibly train a MoE instruction model based on a dense pre-trained model is still an open problem.*

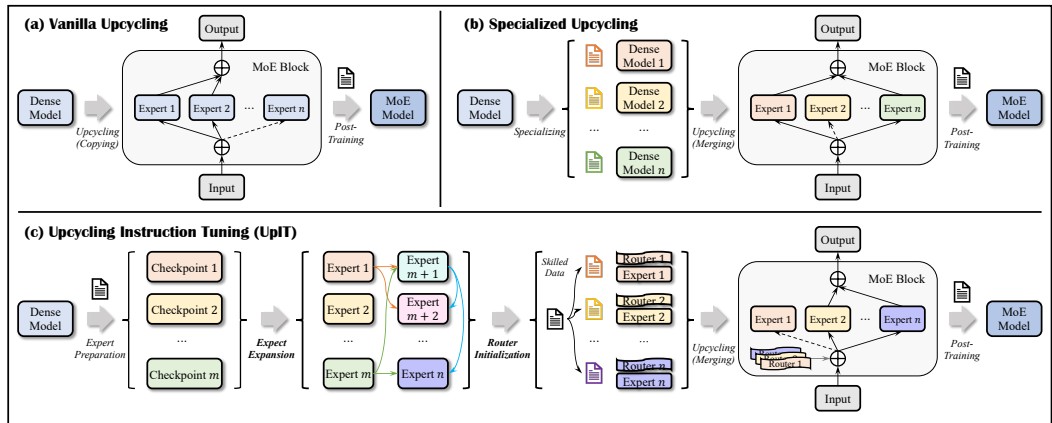

Figure 1: Workflow of vanilla upcycling, specialized upcycling, and the proposed upcycling instruction tuning (UpIT) solutions. UpIT achieves specialized experts with various checkpoints, increases the expert number during the expert expansion stage, and maintains discrepancy among experts through router initialization, thereby achieving efficient and flexible upcycling.

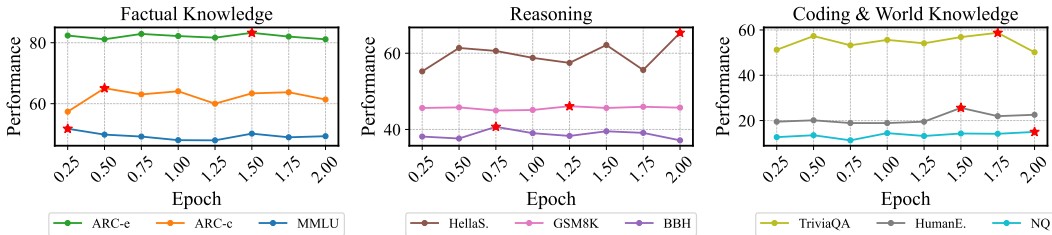

Figure 2: The performance of various checkpoints saved during an instruction tuning process, with a red star indicating the best performance on each benchmark. Checkpoints saved at different epochs excel in different benchmarks, demonstrating the potential as specialized experts.

To answer this question, we first conduct a pilot experiment on dense models to observe the characteristics of models at different epochs during standard instruction tuning. Figure 2 shows checkpoints saved at different epochs exhibit interleaved performance across benchmarks in various domains. In practical terms, we categorize nine benchmarks into four domains: factual knowledge, reasoning, coding, and world knowledge, where the performance on each benchmark generally shows an upward trend followed by a downward trend, but the position of the maximum value varies. For example, the model trained at epoch 2 demonstrates superior performance on HellaSwag and Natual Question, whereas the model trained at epoch 0.25 performs best on MMLU. In other words, models with different training steps demonstrate varying expertise in handling distinct domains. This phenomenon inspires us to consider that *different checkpoints during instruction tuning are inherently suitable for constructing specialized experts*.

In light of the above findings, we propose U̲pcycling I̲nstruction T̲uning (UpIT), which starts from a dense pre-trained model and trains a MoE instruction model with a flexible number of experts, following the basic thought of specialized upcycling. Figure 1 illustrates the four stages of UpIT. Specifically, *(1) Expert Preparation.* Considering the differences among checkpoints in the pilot experiment, it is sufficient to fine-tune the dense pre-trained model and save checkpoints at fixed intervals to prepare for specialized experts, without undertaking meticulous checkpoint selection. *(2) Expert Expansion.* Given the fixed checkpoints, we extend a flexible number of new experts based on genetic algorithms. In each iteration, we select two experts with the greatest differences, merge their parameters and obtain a new expert. We also perform parameter scaling and dropping before merging to simulate the mutation and further promote the discrepancy of experts. *(3) Router Initialization.* Traditional routers are randomly initialized and insensitive to expert capabilities. Here, we assign each expert their skilled data and introduce an auxiliary binary classification loss to pre-

optimize the corresponding routing vector, ensuring that all experts are capable of fully exhibiting their strengths in the MoE model. *(4) Model Upcycling.* Before post-training, we merge the parameters of multiple dense models into a MoE model. Unlike existing methods, the pre-optimized routing vectors are merged into a routing matrix, serving as the final router.

Overall, by leveraging the differences in existing dense checkpoints and introducing the expert expansion stage, UpIT comprehensively reduces the cost of acquiring specialized experts and improves the flexibility of expert numbers. Router initialization further maintains expert diversity, thereby encouraging more effective utilization of data characteristics during the post-training of the MoE model. From an implementation perspective, UpIT divides standard instruction tuning into two parts, where the first part is responsible for expert preparation, and the second is post-training after upcycling. Between these two stages, we find that only 1% of the training data (approximately 500 to 5,000 samples) is enough to pre-optimize the routing vectors, which means that UpIT efficiently upcycles from dense to MoE without significantly increasing the overall data requirements.

We conduct extensive experiments under LoRA-based and FFN-based upcycling settings, with LoRA and FFN as experts. For a fair comparison, we train all models on IDEA dataset (Wu et al., 2024a), considering data sizes ranging from 50K to 500K, and evaluate the performance of nine benchmarks. Experimental results show that UpIT is consistently better under both settings than dense instruction tuning and other upcycling baselines. Especially in situations with small amounts of training data, existing upcycling methods often can not work well, while UpIT utilizes the discrepancy of experts and achieves better results than dense baselines. Moreover, UpIT exhibits excellent scalability, with expected performance improvements when increasing training data, total expert number, or activated expert number. The router visualization and ablation study also verify the overall promoting effect of expert diversity on upcycled MoE models.

In summary, the main contributions of this paper are as follows:

- We propose UpIT, an efficient specialized upcycling method via parameter merging, which can achieve an instruction model with a flexible number of experts. To best of our knowledge, it is the first attempt to utilize intermediate dense checkpoints for model upcycling.

- We emphasize the importance of expert discrepancy in upcycling and incorporate the idea into the entire design of UpIT. The innovative router initialization stage ensures that all specialized experts play to their strengths in the final MoE model.

- Extensive experiments under LoRA and FFN-based settings demonstrate that UpIT significantly outperforms existing methods, whether in scenarios with sufficient or insufficient data, exhibiting outstanding flexibility, scalability, and performance upper bound.

## 2 METHODOLOGY

In this section, we provide a detailed exposition of UpIT. Generally speaking, UpIT perpetuates the concept of specialized upcycling, further using intermediate checkpoints to reduce data requirements, expanding experts to fit the flexible number of experts, and pre-optimizing routing vectors to ensure that each expert in the instruct MoE model leverages their strengths.

### 2.1 PRELIMINARIES

Before introducing UpIT, here we first briefly review some basic concepts of MoE.

**Mixture-of-Experts (MoE).** MoE significantly scale up the total parameter number and increases the knowledge capacity of language models, by selectively activating some of the parameters during inference, it does not proportionally increase the computational workload. In transformers-based models, the feed-forward neural network (FFN) layer in each transformer block is typically replaced with a MoE layer, which comprises $N$ identical and independent experts $\{E_i\}_{i=1}^{N}$, along with a router $R(\cdot)$ for assigning experts, where each expert generally corresponds to an FFN layer, and in the scenario of parameter-efficient fine-tuning (PEFT), the expert may also be LoRA matrices. Formally, for the hidden states $\mathbf{h}$ of the attention layer, the output $\mathbf{o}$ of the MoE layer is represented as $\mathbf{o} = \sum_{i=1}^{N} R(\mathbf{h})E_i(\mathbf{h})$. Here, $E_i(\mathbf{h})$ is the output of the $i$-th expert, $R(\mathbf{h})$ denotes the score for all experts, where experts with the highest scores are usually selected to calculate the final output.

---

**Algorithm 1:** Workflow of UpIT

---

**Input:** Dense pre-trained model $\Theta_d$, training dataset $\mathcal{D}$, target number of experts $n$.

1 // *Expert Preparation*
2 Fine-tune the dense model $\Theta_d$ on $\mathcal{D}$ and obtain a series of checkpoints $\mathcal{C} = \{\Theta_d^1, \ldots, \Theta_d^m\}$.
3 Initialize expert models $\mathcal{E} = \{\mathbf{E}_1, \ldots, \mathbf{E}_m\}$ from checkpoints $\mathcal{C}$.
4 // *Expert Expansion*
5 Merge new expert model with Algorithm 2 and put them into the set $\mathcal{E} = \{\mathbf{E}_1, \ldots, \mathbf{E}_m, \ldots, \mathbf{E}_n\}$.
6 // *Router Initialization*
7 Initialize routing vectors $\mathcal{R} = \{\mathbf{r}_1, \ldots, \mathbf{r}_n\}$ for expert models $\mathcal{E}$.
8 Construct expert-specific data $\mathcal{D}_s = \{\mathcal{D}_s^1, \ldots, \mathcal{D}_s^n\}$ with Algorithm 3.
9 **for** $i = 1$ **to** $n$ **do**
10 $\quad$ Fine-tune expert layer $\mathbf{e}_i$ in $\mathbf{E}_i$ and corresponding routing vector $\mathbf{r}_i$ on $\mathcal{D}_s^i$ with Equation 1.
11 // *Model Upcycling*
12 Initialize router $\mathbf{R}$ by concatenating all routing vectors $\mathcal{R}$.
13 Initialize the MoE model $\Theta_m$ with expert models $\mathcal{E}$ and router $\mathbf{R}$.
14 Fine-tune the MoE model $\Theta_m$ on $\mathcal{D}$.

**Output:** MoE Instruction model $\Theta_m$.

---

**Upcycling.** Upcycling seeks to avoid training MoE models from scratch by transforming an existing dense model into a MoE model, followed by a post-training stage to integrate all the parameters into an organic whole. It starts with dense models, forming experts by expanding the FFN layer or creating new LoRA branches, and then adding routers to control the dispatch of input tokens. In this process, the remaining embedding layers, attention blocks, normalization modules, and output layers are directly transferred from the initial dense model to the ultimate MoE model, and the router is randomly initialized and optimized in the post-training stage.

## 2.2 Workflow of UpIT

Starting from the dense pre-trained model, UpIT achieves a MoE instruction model. Algorithm 1 provides the working sketch. In this section, we provide a detailed explanation of each process.

**Expert Preparation.** In the instruction tuning of LLMs, as shown in Figure 2, the performance of intermediate checkpoints varies across different benchmarks, and different checkpoints exhibit unique strengths in different domains, highlighting the potential to serve as specialized experts. Compared to the labour-intensive method of training diverse expert models with massive domain-specific data (Sukhbaatar et al., 2024), we believe that the natural variations among checkpoints provide a more efficient pathway to developing specialized expert models. By training dense models to generate multiple checkpoints and saving them at regular intervals during training, we can easily obtain a series of expert models proficient in different domains, resulting in a more cost-effective method for preparing specialized expert models.

**Expert Expansion.** Given that the fixed number of checkpoints only sometimes corresponds with the flexible requirements of expert number, acquiring additional checkpoints entails redundant training if the number of experts exceeds the saved checkpoints. Here, we propose to address these challenges by generating distinct experts from existing ones without extensive retraining (see also Algorithm 2). Specifically, we draw inspiration from genetic algorithms, where two experts with the greatest differences are selected as parents in each iteration. We simulate the mutation process by randomly assigning weights to the parents and apply DARE (Yu et al., 2024) to introduce mutations into the newly created expert further, enhancing its discrepancy and adaptability. Such an expansion process not only eliminates the need for additional retraining but also facilitates the flexible expansion of the number of experts, ultimately improving the scalability of UpIT.

**Router Initialization.** Since routers remain randomly initialized after upcycling, which leads to the misallocation of tokens in the early post-training stages, in UpIT, such misallocation will weaken the expert differences in the previous stage and impact the learning efficiency of MoE models. To solve this problem, we propose a data selection approach to curate expert-specific data tailored to each expert model and pre-optimize additional routing vectors to ensure the discrepancy among experts (see also Algorithm 3). Specifically, we initially embed one-dimensional routing vectors $\mathcal{R}$ before

---

**Algorithm 2:** Genetic Algorithm-Based Expert Expansion

---

**Input:** Existing $m$ expert models $\mathcal{E} = \{\mathbf{E}_1, \cdots, \mathbf{E}_m\}$, target number of experts $n$.

1 **for** $i = 1$ **to** $n - m$ **do**
2   **for** $j = 1$ **to** $len(\mathcal{E})$ **do**
3     **for** $k = j + 1$ **to** $len(\mathcal{E})$ **do**
4       Find two expert models $\mathbf{E}_{j*}$, $\mathbf{E}_{k*}$ with the smallest similarity.
5   Setting weights $\alpha$ and $\beta$ randomly, s.t., $\alpha + \beta = 1$.
6   Merge new expert model via $\mathbf{E}_{m+i} = \text{DARE}(\alpha\mathbf{E}_{j*}, \beta\mathbf{E}_{k*})$ and put $\mathbf{E}_{m+i}$ into $\mathcal{E}$.

**Output:** Expanded expert models $\mathcal{E} = \{\mathbf{E}_1, \cdots, \mathbf{E}_m, \cdots, \mathbf{E}_n\}$.

---

**Algorithm 3:** Expert-Specific Data Selection for Router Initialization

---

**Input:** Training dataset $\mathcal{D}$, Expert models $\mathcal{E} = \{\mathbf{E}_1, \cdots, \mathbf{E}_n\}$, and data capacity for each expert $C$.

1 Initialize $n$ empty expert-specific data buckets $\{\mathcal{D}_s^1, \ldots, \mathcal{D}_s^n\}$.
2 Construct seed dataset $\mathcal{D}_s$ by randomly selecting 1% of the data in $\mathcal{D}$.
3 **foreach** data $d_i$ in $\mathcal{D}_s$ **do**
4   Calculate the perplexity list $\mathcal{P}_i = [p_i^1, \cdots, p_i^n]$ of each expert on $d_i$.
5   Sort $\mathcal{P}_i$ in order (with small perplexity at the beginning), denoted as $\mathcal{P}_i'$.
6   **foreach** $p_i^j$ in $\mathcal{P}_i'$ **do**
7     Get the expert index $j$.
8     **if** $len(\mathcal{D}_e^j) < C$ **then**
9       Append data $d_i$ to bucket $\mathcal{D}_e^j$.
10      Break.

**Output:** Expert-specific data buckets $\{\mathcal{D}_s^1, \ldots, \mathcal{D}_s^n\}$.

---

the MoE layer in each transformer block and participate in the training process as expert-specific routers. Next, we introduce an auxiliary loss $\mathcal{L}_{aux}$ intending to maximize the output probability of corresponding routing vectors. This ensures that the likelihood of tokens being assigned to appropriate experts increases when they pass through the router. The pre-optimizing objective of $i$-th expert model is formulated as follows,

$$\mathcal{O}_i = \min_{\mathbf{E}_i}(\alpha\mathcal{L}_{lm}(\mathbf{E}_i) + (1 - \alpha)\mathcal{L}_{aux}(\mathbf{E}_i)) \tag{1}$$

where $\alpha$ is the balance coefficient, which we set to 0.5 in our experiments, and $\mathcal{L}_{lm}(\cdot)$ is the causal language model loss. The auxiliary loss $\mathcal{L}_{aux}(\cdot)$ is defined as follows,

$$\mathcal{L}_{aux}(\mathbf{E}_i) = \text{CrossEntropy}(\text{Sigmoid}(\mathbf{h}_{\mathbf{r}_i}), \mathbf{I}) \tag{2}$$

where $\mathbf{h}_{\mathbf{r}_i}$ is the output of routing vector and $\mathbf{I}$ is the unit matrix. We use Sigmoid function to scale the output to $(0, 1)$ and minimize its difference from $\mathbf{I}$, which is equivalent to maximizing the output probability of the routing vector on the data that current expert model excels at.

**Model Upcycling.** Finally, we upcycle the dense model $\Theta_d$ to MoE model $\Theta_m$ by merging all the expert models $\mathcal{E}$ and routing vectors $\mathcal{R}$. Specifically, for the initialization of experts, we utilize pre-optimized expert models from $\mathcal{E}$. In terms of router initialization, we concatenate all routing vectors from $\mathcal{R}$ to form a complete router $\mathbf{R} \in \mathbb{R}^{d_h, n}$, where $d_h$ is the dimension of hidden states, This way, the obtained MoE block could allocate different tokens to experts skilled in processing them. Finally, we continue to utilize $\mathcal{D}$ for post-training to achieve the final MoE model.

### 2.3 TRAINING DETAILS

To comprehensively evaluate the effectiveness of UpIT, we utilize two types of upcycling settings:

(1) **FFN-based Upcycling:** Initially, we fully fine-tune all parameters of the dense pre-trained model to accumulate several expert models. In the expert expansion stage, we apply the genetic algorithm to construct expert modules (i.e. FFN layers), average the parameters of backbone modules (i.e. all

Table 1: Performance comparison under Lora-based and FFN-based upcycling settings, where (`xE,Ay`) indicates that `y` out of `x` experts are activated, Lora-based UpIT (`16E,A2`) and FFN-based UpIT (`8E,A2`) are expanded from Lora-based UpIT (`8E,A2`) and FFN-based UpIT (`4E,A2`), respectively. Bold text and underlined text denote the best and second-best results in each group.

| | HumanE. | GSM8K | HellaS. | BBH | MMLU | NQ | Tri.QA | ARC-c | ARC-e | Avg. |
|---|---|---|---|---|---|---|---|---|---|---|
| *LoRA-based Models* | | | | | | | | | | |
| Llama 2 7B | 14.63 | 13.95 | 26.58 | 34.73 | 39.84 | 10.06 | 62.06 | 37.29 | 50.26 | 32.16 |
| LoRA | 22.56 | 45.72 | 65.36 | 37.14 | 49.33 | 14.99 | 50.15 | 61.36 | 81.13 | 47.53 |
| Self-MoE (`8E,A2`) | 28.05 | 46.70 | 64.27 | 38.67 | 49.63 | 21.11 | 48.67 | 64.41 | 82.19 | 49.30 |
| PESC (`8E,A2`) | 28.05 | 46.55 | 63.14 | 37.59 | 46.12 | 16.68 | 49.58 | 61.36 | 74.60 | 47.07 |
| LoRAMoE$_\text{PT}$ (`8E,A2`) | 34.15 | 47.61 | 60.89 | 37.40 | 46.61 | 17.62 | 46.33 | 60.68 | 74.60 | 47.32 |
| LoRAMoE$_\text{SFT}$ (`8E,A2`) | 28.66 | **49.81** | **67.62** | 38.88 | **50.54** | 20.55 | 50.16 | 62.37 | 81.31 | 49.99 |
| UpIT (`8E,A2`) | **35.37** | 49.51 | 66.00 | **40.27** | 50.31 | **24.52** | 55.27 | 65.08 | **83.60** | **52.21** |
| Self-MoE (`16E,A2`) | 30.20 | 47.61 | 65.36 | 37.14 | 49.33 | 24.52 | 51.11 | 62.71 | 82.19 | 50.02 |
| PESC (`16E,A2`) | 31.10 | 47.62 | 63.14 | 37.59 | 49.08 | 20.83 | 49.58 | 63.05 | 77.62 | 48.85 |
| LoRAMoE$_\text{PT}$ (`16E,A2`) | 40.24 | 46.55 | 65.89 | 36.39 | 48.53 | 19.36 | 46.19 | 61.69 | 76.01 | 48.98 |
| LoRAMoE$_\text{SFT}$ (`16E,A2`) | 30.12 | **49.62** | **66.77** | 40.21 | **50.96** | 20.83 | 52.63 | 63.41 | 80.67 | 50.58 |
| UpIT (`16E,A2`) | **40.62** | 48.37 | 66.62 | 39.43 | 50.70 | **25.62** | 56.61 | 67.46 | **84.66** | **53.34** |
| *FFN-based Models* | | | | | | | | | | |
| Sheared Llama 2.7B | 5.49 | 1.74 | 25.09 | 26.62 | 26.98 | 6.43 | 38.89 | 22.37 | 24.69 | 19.81 |
| SFT | 26.22 | 29.19 | 38.01 | 26.46 | 33.93 | 8.42 | 18.61 | 42.37 | 58.55 | 31.31 |
| Self-MoE (`4E,A2`) | 6.71 | 8.87 | 32.11 | 27.65 | 28.81 | **18.45** | **42.27** | 33.22 | 47.44 | 27.28 |
| Upcycle$_\text{PT}$ (`4E,A2`) | **31.71** | **35.10** | 43.40 | **30.23** | 37.93 | 13.74 | 34.72 | 45.08 | 58.73 | 36.74 |
| Upcycle$_\text{SFT}$ (`4E,A2`) | 23.17 | 33.97 | **50.27** | 29.50 | **38.90** | 15.18 | 34.20 | **48.14** | 65.08 | 37.60 |
| UpIT (`4E,A2`) | 31.34 | 33.81 | 48.97 | 29.53 | 40.84 | 14.71 | 36.99 | 47.80 | 65.96 | **38.88** |
| Self-MoE (`8E,A2`) | 10.62 | 22.73 | 34.69 | 28.95 | 30.10 | **15.68** | 37.68 | 40.00 | 50.37 | 30.09 |
| Upcycle$_\text{PT}$ (`8E,A2`) | 26.22 | 34.04 | **51.57** | 28.95 | 39.84 | 13.57 | 33.86 | 53.22 | 66.49 | 38.64 |
| Upcycle$_\text{SFT}$ (`8E,A2`) | 22.56 | 33.66 | 46.26 | 29.25 | 39.19 | 14.76 | 35.18 | **49.15** | 67.72 | 37.53 |
| UpIT (`8E,A2`) | **32.19** | **35.64** | 49.15 | **30.23** | **40.38** | 14.57 | 37.93 | 49.10 | **68.43** | **39.74** |

layers except FFN) in candidate expert models, and result in new diverse expert models. We select expert-specific data to pre-optimize the FFN layers and routing vectors during the router initialisation stage. Finally, in the model upcycling stage, we average the backbone parameters of all expert models and concatenate the routing vectors, integrating FFN layers to produce the final MoE models.

(2) **LoRA-based Upcycling:** The key difference from FFN-based upcycling is that in parameter-efficient fine-tuning, parameters of backbone modules remain unchanged. Instead, we augment FFN layers with LoRA matrices, and operate on the values of LoRA matrices during expert expansion.

Following previous work (Fedus et al., 2022), during post-training, we also use load balancing loss, $\mathcal{L}_\text{load} = n \cdot \sum_{i=1}^{n} f_i \cdot P_i$, where $n$ is the expert number, $f_i$ is the fraction of tokens dispatched to expert $E_i$, $P_i$ is the average fraction of the router probability allocated for expert $E_i$.

## 3 EXPERIMENTS

### 3.1 EXPERIMENTAL SETUP

**Baselines.** To assess the effectiveness of UpIT, we compare its performance against several baselines. For LoRA-based settings, we consider the following baselines. (1) LoRA (Hu et al., 2021), (2) Self-MoE (Kang et al., 2024), (3) PESC (Wu et al., 2024a), (4) LoRAMoE$_\text{PT}$ (Dou et al., 2024), and (5) LoRAMoE$_\text{SFT}$. For FFN-based settings, we compare UpIT with (1) SFT, (2) Self-MoE (Kang et al., 2024), (3) Upcycle$_\text{PT}$ (Komatsuzaki et al., 2023), and (4) Upcycle$_\text{SFT}$. For a more detailed description of the baselines, please refer to Appendix A.1.

**Dataset.** Following (Wu et al., 2024a), we simultaneously train UpIT and compared baselines on a diverse set of datasets, encompassing Magicoder (Wei et al., 2023) for coding, MetaMathQA (Yu et al., 2023) for mathematical and SlimORCA (Lian et al., 2023) for other general abilities from various subjects. We do not perform quality filtering or other operations on the three datasets. We randomly sample data in a 1:1:3 ratio to create the final training dataset with 500K samples.

**Implementation Details.** We utilize Llama 2 7B and Sheared Llama 2.7B to train LoRA-based and FFN-based models. We adopt a constant learning rate of 2e-4 and 2e-5 for LoRA-based and

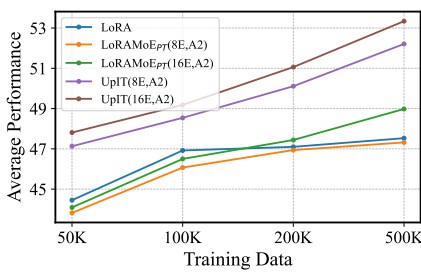 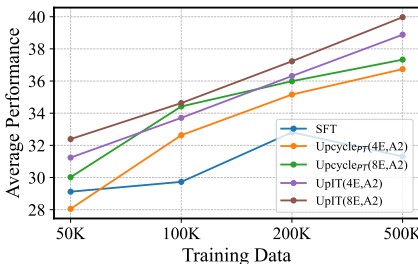

Figure 3: Performance comparison of UpIT and vanilla upcycling methods under different size of training data. Detailed results in Section A.3

FFN-based settings, respectively. All models are trained for 4 epochs in total. For UpIT, we first train the dense model for 2 epochs and prepare specialized expert models using intermediate checkpoints saved between epochs 1 and 2. In router initialization, we randomly select 1% of the training data and train for 4 epochs to pre-optimize routing vectors. In model upcycling, UpIT does not introduce additional training by training the upcycling MoE models for 2 epochs. Therefore, the total training duration also amounts to 4 epochs, including 2 epochs for expert preparation and 2 for post-training. All experiments are conducted using a global batch size of 128 and a context length of 2048, running on 8 NVIDIA A800 GPUs. For evaluation, we use various benchmarks to validate comprehensively our method. Please refer to Appendix A.2 for detailed evaluation benchmarks and metrics.

## 3.2 MAIN RESULTS

Table 1 showcases a comparative analysis of benchmark results for LoRA-based and FFN-based models across diverse domains, revealing the following key insights.

(1) The proposed UpIT framework demonstrates remarkable performance across various benchmarks, highlighting its effectiveness compared to existing upcycle solutions. Specifically, when compared with LoRAMoE$_{SFT}$(8E,A2), the LoRA-based UpIT(8E,A2) achieves an average performance improvement of 2.22%. When the number of experts is expanded to 16, UpIT(16E,A2) sustains a competitive edge over LoRAMoE$_{SFT}$(16E,A2), exhibiting a lead of 2.76%. Similar trends are observed in FFN-based scenarios, where UpIT(4E,A2) and UpIT(8E,A2) outperform Upcycle$_{SFT}$(4E,A2) and Upcycle$_{SFT}$(8E,A2) by 1.28% and 2.21%, respectively. This comprehensive analysis further corroborates the applicability of UpIT across diverse MoE architectures, consistently yielding optimal performance.

(2) In comparisons with PESC(8E,A2) and PESC(16E,A2), which take adapter structure as experts, LoRAMoE$_{PT}$(8E,A2) and LoRAMoE$_{PT}$(16E,A2) display respective advantages of 0.25% and 0.13%, thereby underscoring a slightly superiority of LoRA-based MoE models over adapter-based counterparts. More than that, in FFN-based upcycling, two Self-MoE models experience a collapse in performance, a phenomenon not observed in LoRA-based settings. We posit that this is due to the excessive number of expert parameters introduced in FFN-based upcycling, and the small data in instruction tuning is insufficient to differentiate the experts sufficiently, which hinders the ability of only training routers to fit the diverse data effectively.

## 3.3 SCALING THE TRAINING DATASET

To assess the data-efficient nature of UpIT, we validate UpIT and vanilla upcycling approaches by randomly sampling 50K, 100K, and 200K samples from the full 500K dataset, enabling experiments across four data sizes. As illustrated in Figure 3, we have several intriguing findings.

(1) In the context of LoRA-based scenarios, UpIT(8E,A2) demonstrates comparable performance to LoRAMoE(8E,A2) trained on 500K samples, with only 50K training samples. When scaling up to 16 experts, UpIT(16E,A2) outperforms LoRAMoE(16E,A2) trained on the full 500K dataset again with 100K training samples. These findings extend to FFN-based settings, underscoring the data-efficient essence of UpIT and the ability to diminish the data demands of upcycling notably.

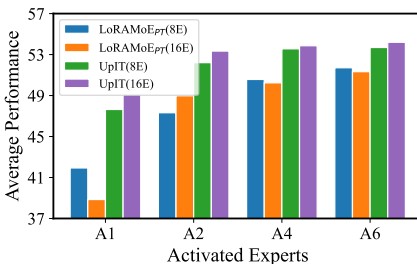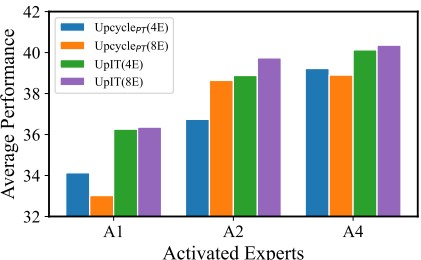

Figure 4: Performance comparison of UpIT and vanilla upcycling methods under different total and activated experts. Detailed results in Section A.4.

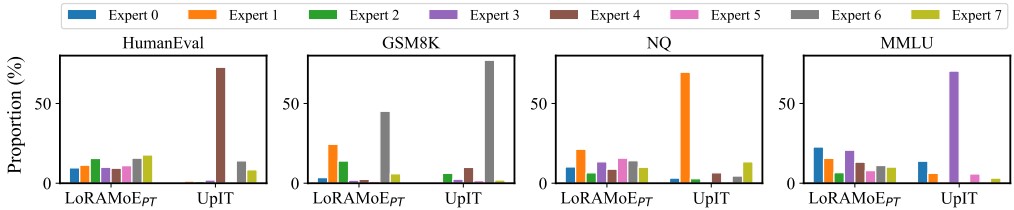

Figure 5: Proportion of tokens dispatched to each expert on different benchmarks, where experts in UpIT exhibit stronger diversity than LoRAMoE.

(2) Both existing LoRA-based and FFN-based models face performance growth saturation issues with traditional SFT (LoRA) and vanilla upcycling strategies. Specifically, while a noticeable performance increase occurs as the dataset scales from 50K to 200K, the performance growth stabilizes as it continues to expand from 200K to 500K, with the average performance exhibiting a log-like curve. In contrast, UpIT demonstrates nearly linear growth trends in FFN-based models and even exhibits accelerated performance gains as the dataset size increases in LoRA-based models. This strongly indicates that MoE models trained using UpIT could efficiently capture the principles of token dispatch more and possess a higher performance upper bound.

## 3.4 SCALING THE NUMBER OF EXPERTS

To examine the impact of scaling the number of total experts and activated experts on MoE models, we investigate the effects of UpIT and vanilla upcycling methods with different expert numbers. The first conclusion drawn from Figure 4 is that UpIT demonstrates superior performance across all configurations. Furthermore, as the number of activated experts increases, the growth trend of average performance gradually slows down, which is attributed to the fact that the evaluation benchmark is domain-specific, and simply increasing the number of activated parameters does not consistently yield substantial improvements. We also find that under the same activated parameters, as the number of experts increases, vanilla upcycling even experiences several performance drops, whereas UpIT consistently shows improvements in performance as the number of experts grows. Due to the inefficiency of data utilization in vanilla upcycling, increasing the number of experts during training leads to a reduction in data allocation for each expert, and the router fails to dispatch tokens to experts appropriately, results in unpredictable model performance.

## 3.5 ROUTER ANALYSIS

To assess the efficiency and interoperability of UpIT, understanding its token dispatch mechanism is essential. We comprehensively analyze the distribution patterns of designated experts across four representative benchmarks: HumanEval, GSM8K, NQ, and MMLU. The results of this examination are illustrated in Figure 5, focusing specifically on the 15th layer of LoRAMoE and UpIT with 8 total experts and 2 activated experts. Significantly, Expert 4 exhibits significantly higher activation within the HumanEval benchmark compared to the other datasets, while Expert 3 demonstrates a substantial activation rate in MMLU compared to other experts. The analysis reveals that, aside from exhibiting

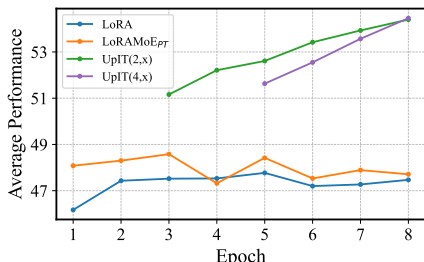 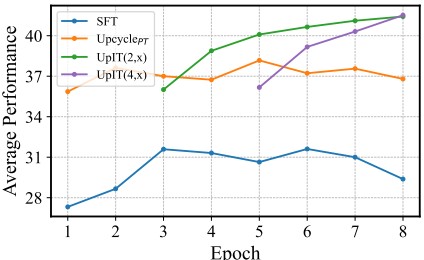

Figure 6: Performance comparison of UpIT and vanilla upcycling methods with different training epochs where x represent training epochs. For UpIT, we consider different allocations of the training epochs between expert preparation and model upcycling. Detailed results are shown in Section A.5

|  | Avg. |  | Avg. |  | Avg. |
|---|---|---|---|---|---|
| UpIT(Front.Half) | 51.37 | UpIT(w/o EE) | 53.31 | UpIT(w/o Init) | 49.96 |
| UpIT(Uniform) | 51.71 | UpIT(Random) | 52.41 | UpIT(Random) | 49.30 |
| UpIT(Back.Half) | **52.21** | UpIT(Genetic) | **53.34** | UpIT(Skilled) | **52.21** |

Table 2: Further analysis containing different checkpoint selection approaches (left), different expert expansion methods (middle), and different router initialization strategies (right). Detailed results are shown in Section A.6.1, A.6.2, and A.6.3

slight routing preferences in GSM8K, LoRAMoE dispatches tokens evenly among the experts across the other three benchmarks. In contrast, UpIT accurately allocates tokens from different domains to specific experts, highlighting the significant differentiation among routers and experts, resulting in a more data-efficient model upcycling routine.

## 3.6 EXPLORE THE UPPER BOUND OF UpIT

In the main experiment, we aligned total training amounts fairly to compare UpIT with baseline methods. Here, we extend the training epochs to better understand the performance upper bound of UpIT. For the baselines, we continue training for an additional 4 epochs, but they do not show significant performance gains. Instead, we expand the training for UpIT in two ways: UpIT(2,6) includes 2 epochs for expert preparation followed by 6 epochs for post-training, while UpIT(4,4) comprises 4 epochs for expert preparation followed by 4 epochs for post-training. Figure 6 shows that UpIT demonstrates continuous performance improvement with more training epochs, indicating greater potential than the baselines. Besides, it is interesting that UpIT(4,4) experiences longer iteration epochs during expert preparation, with greater divergences between expert models, leading to a more rapid upward trend. In the post-training stage, after only 4 epochs, it achieves comparable results to UpIT(2,6) and is expected to reach higher performance in the upper bound.

## 3.7 FURTHER ANALYSIS

**Different Checkpoint Selection Strategies during Expert Preparation.** In this section, we would like to answer the question of how to select checkpoints if the number of saved checkpoints exceeds the required number of experts. As shown in Table 2 (left), the latter-half selection performs better. Detailed results in Table 8 indicate that the primary performance differences stem from improvements in mathematical reasoning and coding abilities as training progresses, and selecting later checkpoints might enhance these capabilities.

**Different Parameter Merging Strategies during Expert Expansion.** Next, we investigate various expert merging strategies. Table 2 (middle) reveals that the genetic algorithm-based expert expansion achieves performance comparable to the method of constructing experts with more checkpoints, highlighting the effectiveness of our merging strategy in generating diverse experts. Additionally, merging two randomly selected expert models in each round results in a performance decline, again validating the importance of maintaining expert diversity.

**Different Data Selection Strategies during Router Initialization.** We compare the expert-specific data selection approach with two alternatives: without the router initialization stage and randomly selecting data to pre-optimize routing vectors. Table 2 (right) illustrates a significant performance decline that occurs without router initialization, underscoring the importance of this stage and the effectiveness of utilizing checkpoints as experts. Furthermore, performance diminishes when the training data is randomly selected, due to the loss of inherent diversity among checkpoints when using similar data in pre-optimizing different expert models. Overall, the phenomenon in this paper is similar to the findings of discussing expert diversity in existing work (Lo et al., 2024)

## 4 RELATED WORK

**Mixture of Experts.** Mixture of Experts (MoE) (Jacobs et al., 1991) modifies the FFN layers or inserts additional branches to construct experts and activates them sparsely, thereby significantly enlarging the model capacity without noticeably increasing computational costs. The exploration of MoE has increasingly captured attention in recent years. Vanilla upcycling (Komatsuzaki et al., 2023) copies FFN layers, followed by post-training, have achieved a more convenient MoE training strategy. LoRAMoE (Dou et al., 2024), MoELoRA (Luo et al., 2024), MixLoRA (Li et al., 2024b) and MoLE (Wu et al., 2024b) develop an MoE model by incorporating several LoRA branches as experts, utilizing sparse activation or linear weighting for model construction. PESC (Wu et al., 2024a) introduces adapter-based structures after the FFN layers, exploring a parameter-efficient MoE model that diverges from the LoRA paradigm. MoExtend (Zhong et al., 2024) adapt to new tasks by expanding the MoE layer during the training process, mitigating catastrophic forgetting. MoE Jetpack (Zhu et al., 2024b) introduces checkpoint recycling, which leverages checkpoints to enhance the flexibility and diversity of expert initialization. In contrast, Branch-Train-MiX (Sukhbaatar et al., 2024) and Self-MoE (Kang et al., 2024) explore a Specialized Upcycling method by introducing specialized experts. Despite superior performance, they still require considerable domain data to acquire specialized experts. In this paper, we integrate the advantages of the work above and utilize intermediate checkpoints for expert preparation, innovatively propose an expert expansion strategy and an stage of pre-optimizing routing vectors to enhance flexibility, scalability and data efficiency.

**Model Merging.** Model merging has emerged as a prominent research direction in recent years, focusing on consolidating multiple task-specific models into a unified model with diverse capabilities (Wortsman et al., 2022; Ilharco et al., 2023). Model merging usually considers the combination of model parameters without accessing the original training data. Average Merging (Wortsman et al., 2022) is one typical model merging approach, which utilizes averaged parameters to construct the merged model. Task Arithmetic (Zhang et al., 2023) employs a pre-defined scaling term to distinguish the importance of various models. Fisher Merging (Matena & Raffel, 2022) performs automatic weighted merging of parameters, where the Fisher information matrix calculates the weights. TIES-Merging (Yadav et al., 2023) tackles the task conflicts in (Zhang et al., 2023) by trimming low-magnitude parameters, resolving sign disagreements, and disjointly merging parameters with consistent signs. DARE (Yu et al., 2024) first sparsifies delta parameters of several SFT homologous models and then merges them into a single model. We innovatively integrate the model merging concept into the MoE model, leveraging genetic algorithms and DARE to expand and evolve new experts. This approach enhances the scalability of our framework.

## 5 CONCLUSION

In this paper, we present a novel, flexible, scalable, and data-efficient approach, Upcycling Instruction Tuning (UpIT), for transforming dense pre-trained models into MoE instruction models. By leveraging intermediate checkpoints as specialized experts and implementing an expert expansion stage with genetic algorithms and parameter merging, UpIT successfully enhances expert diversity while allowing for a flexible number of experts. The strategic selection of seed data ensures that each expert and router performs optimally within the MoE framework. Our extensive experiments demonstrate that UpIT not only achieves superior performance across various benchmarks but also maintains stability in expert and data scaling. Further analysis emphasizes the critical importance of expert diversity in the upcycling process. Overall, UpIT offers a promising pathway for enhancing the efficiency and effectiveness of MoE models, paving the way for future advancements in the field.

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

# A    APPENDIX

## A.1    DETAILED DESCRIPTION OF BASELINES

For LoRA-based settings, we compare several baselines with our proposed UpIT. (1) LoRA (Hu et al., 2021): It adds low-rank matrix branches for parameter-efficient fine-tuning. (2) Self-MoE (Kang et al., 2024): It employs specialized experts to build MoE model, and only train the routers during the post-training stage. We solely reuse the training strategy of Self-MoE, and only train the routers after upcycling with intermediate checkpoints. (3) PESC (Wu et al., 2024a): It adds several adapter structures after the FFN block as experts. (4) LoRAMoE$_{PT}$ (Dou et al., 2024): It employs the same structure as UpIT which insert several LoRA branches as experts, and (5) LoRAMoE$_{SFT}$: It copies the final-step checkpoint to form MoE blocks.

Similarly, for FFN-based settings, we compare UpIT with (1) SFT, It is the standard fine-tuning solution. (2) Self-MoE (Kang et al., 2024), It is similar with the LoRA-based method. (3) Upcycle$_{PT}$ (Komatsuzaki et al., 2023): It is the vanilla upcycling approach for transforming a dense pre-trained model to the MoE model. (4) Upcycle$_{SFT}$. It copies the final-step checkpoint to form MoE blocks.

## A.2    EVALUATION METRICS.

To validate the effectiveness of our method, we employ comprehensive evaluation benchmarks, which contain various abilities. (1) **Factual Knowledge**: To assess the LLMs' factual knowledge, we employ the Massive Multitask Language Understanding (MMLU) (Hendrycks et al., 2021), ARC-e and ARC-c (Clark et al., 2018) datasets. MMLU comprises questions across 57 subjects from elementary to professional difficulty levels. ARC-e and ARC-c contain questions for science exams from grade 3 to grade 9. We report the 0-shot accuracy based on answer perplexity for MMLU and ARC. (2) **Reasoning**: We utilize the test split of the Grade School Math (GSM8K) (Cobbe et al., 2021), HellaSwag (HellaS.) (Zellers et al., 2019) and Big-Bench-Hard (BBH) (Suzgun et al., 2022) benchmarks to evaluate reasoning abilities. We report the 8-shot accuracy for GSM8K and the 3-shot accuracy for HellaSwag. (3) **Coding**: To probe the LLMs' ability to generate functionally correct programs from docstrings, we utilize HumanEval (HumanE.) (Chen et al., 2021) and report the pass@1 performance. (4) **World Knowledge**: We adopt Natural Question (NQ) (Kwiatkowski et al., 2019) and TriviaQA (Joshi et al., 2017) to evaluate the commonsense question-answering ability. All of the above evaluations are performed using opencompass (Contributors, 2023) framework, and to expedite evaluation, we enable batch-padding with a batch size of 32.

## A.3    DETAIL RESULTS OF SCALING THE TRAINING DATASET

Table 3 and 4 presents the detailed results of LoRA-based and FFN-based models under 50K, 100K, 200K and 500K training samples which correspond to Figure 3. The detailed results further substantiate our findings, demonstrating that UpIT exhibits higher data utilization efficiency. It achieves stronger performance with less data, showcasing the data-efficient nature of UpIT.

## A.4    DETAIL RESULTS OF SCALING THE EXPERTS

Table 5 shows the detailed results of different numbers of experts and different activated parameters which correspond to Figure 4. The detailed results indicate that UpIT consistently maintains the desired growth trend during the scaling of experts and activated parameters, whereas the baselines exhibit an unstable performance growth trend during scaling, making it difficult to reliably predict performance expectations.

## A.5    DETAILED RESULTS OF UPPER BOUND

Table 6 and 7 show the detailed results of performance upper bound during continuous training for LoRA-based and FFN-based models, respectively. The detailed results demonstrate that UpIT maintains a consistent trend of gradual performance improvement during ongoing training, while SFT (LoRA) and Upcycle$_{PT}$ (LoRAMoE) exhibit performance instability throughout this process. Furthermore, compared to UpIT(2), UpIT(4) benefits from a broader selection range of expert

Table 3: Detailed results of LoRA-based models under four data sizes, where (xE,Ay) indicates that y out of x experts are activated. LoRA-based UpIT(16E,A2) is expanded from LoRA-based UpIT(8E,A2). Bold text and underlined text denote the best and second-best results in each group.

| | HumanE. | GSM8K | HellaS. | BBH | MMLU | NQ | TriviaQA | ARC-c | ARC-e | Avg. |
|---|---|---|---|---|---|---|---|---|---|---|
| *Results of 50K training samples* | | | | | | | | | | |
| LoRA | 18.29 | **37.00** | 51.84 | 39.71 | **48.61** | 15.46 | 56.27 | 55.93 | 76.90 | 44.45 |
| LoRAMoE$_{PT}$(8E,A2) | 17.07 | 34.87 | 52.21 | 37.33 | 47.24 | 22.66 | 53.64 | 54.24 | 75.13 | 43.82 |
| LoRAMoE$_{PT}$(16E,A2) | **21.34** | 35.48 | 46.01 | 38.39 | 47.97 | 23.38 | 52.69 | 56.95 | 74.60 | 44.09 |
| UpIT(8E,A2) | 18.90 | 33.74 | 58.19 | **39.95** | 47.26 | 27.04 | 60.42 | 58.64 | 80.07 | 47.13 |
| UpIT(16E,A2) | 18.63 | 35.67 | 60.11 | 39.87 | 47.35 | 26.59 | **61.37** | 59.96 | 80.76 | **47.81** |
| *Results of 100K training samples* | | | | | | | | | | |
| LoRA | **21.95** | **40.94** | 60.50 | 37.32 | 50.97 | 14.88 | 55.41 | 62.37 | 77.95 | 46.92 |
| LoRAMoE$_{PT}$(8E,A2) | 18.90 | 36.54 | 57.06 | 37.84 | 48.93 | 22.99 | 54.05 | 60.00 | 78.31 | 46.07 |
| LoRAMoE$_{PT}$(16E,A2) | 20.12 | 39.73 | 58.31 | 36.50 | 47.49 | 23.10 | 53.62 | 62.03 | 77.60 | 46.50 |
| UpIT(8E,A2) | 18.90 | 39.58 | 60.01 | 38.17 | 51.18 | 25.79 | 58.76 | 64.41 | 80.07 | 48.54 |
| UpIT(16E,A2) | 20.12 | 40.41 | 61.50 | 39.73 | 50.43 | 26.59 | 59.35 | 63.73 | 80.78 | 49.18 |
| *Results of 200K training samples* | | | | | | | | | | |
| LoRA | 20.73 | **46.32** | 58.86 | 38.64 | 48.79 | 16.07 | 53.93 | 61.69 | 78.84 | 47.10 |
| LoRAMoE$_{PT}$(8E,A2) | 26.22 | 42.23 | 58.36 | 37.63 | 47.07 | 20.97 | 51.79 | 60.75 | 77.45 | 46.94 |
| LoRAMoE$_{PT}$(16E,A2) | 30.49 | 40.03 | 60.59 | 36.67 | 48.42 | 21.52 | 50.80 | 62.03 | 76.37 | 47.44 |
| UpIT(8E,A2) | 20.12 | 43.97 | 65.16 | 40.46 | 51.18 | 25.54 | 58.54 | 66.10 | 79.89 | 50.11 |
| UpIT(16E,A2) | 22.82 | 44.96 | 64.89 | 40.64 | 51.02 | 25.69 | 60.26 | 65.97 | 83.25 | 51.06 |
| *Results of 500K training samples* | | | | | | | | | | |
| LoRA | 22.56 | 45.72 | 65.36 | 37.14 | 49.33 | 14.99 | 50.15 | 61.36 | 81.13 | 47.53 |
| LoRAMoE$_{PT}$(8E,A2) | 34.15 | 47.61 | 60.89 | 37.40 | 46.61 | 17.62 | 46.33 | 60.68 | 74.60 | 47.32 |
| LoRAMoE$_{PT}$(16E,A2) | 40.24 | 46.55 | 65.89 | 36.39 | 48.53 | 19.36 | 46.19 | 61.69 | 76.01 | 48.98 |
| UpIT(8E,A2) | 35.37 | 49.51 | 66.00 | 40.27 | 50.31 | 24.52 | 55.27 | 65.08 | 83.60 | 52.21 |
| UpIT(16E,A2) | 40.62 | 48.37 | 66.62 | 39.43 | 50.70 | 25.62 | 56.61 | 67.46 | 84.66 | 53.34 |

Table 4: Detailed results of FFN-based models under four data sizes, where (xE,Ay) indicates that y out of x experts are activated. FFN-based UpIT(8E,A2) is expanded from FFN-based UpIT(4E,A2). Bold text and underlined text denote the best and second-best results in each group.

| | HumanE. | GSM8K | HellaS. | BBH | MMLU | NQ | TriviaQA | ARC-c | ARC-e | Avg. |
|---|---|---|---|---|---|---|---|---|---|---|
| *Results of 50K training samples* | | | | | | | | | | |
| SFT | **9.76** | 12.28 | 34.43 | 28.43 | 29.57 | **16.01** | **40.13** | 40.00 | 51.50 | 29.12 |
| Upcycle$_{PT}$(4E,A2) | 9.15 | 15.95 | 29.90 | 29.20 | 31.75 | 14.46 | 32.57 | 38.64 | 50.79 | 28.05 |
| Upcycle$_{PT}$(8E,A2) | 7.93 | 13.87 | 38.34 | 29.23 | 33.60 | 14.40 | 35.68 | 41.36 | 55.73 | 30.02 |
| UpIT(4E,A2) | 8.54 | 14.78 | 33.05 | 28.39 | 35.72 | 14.54 | 37.68 | 44.41 | 64.02 | 31.24 |
| UpIT(8E,A2) | 9.62 | **17.39** | 39.29 | 29.20 | 36.12 | 14.35 | 37.93 | 44.41 | 63.20 | **32.39** |
| *Results of 100K training samples* | | | | | | | | | | |
| SFT | 6.71 | 13.27 | 35.87 | 28.55 | 30.95 | **15.68** | **40.75** | 41.49 | 54.32 | 29.73 |
| Upcycle$_{PT}$(4E,A2) | 14.02 | 20.17 | 42.48 | **29.44** | 36.12 | 14.07 | 33.41 | 45.08 | 58.91 | 32.63 |
| Upcycle$_{PT}$(8E,A2) | 12.20 | 21.08 | 47.75 | 28.90 | 38.53 | 14.07 | 35.28 | 48.81 | 63.14 | 34.42 |
| UpIT(4E,A2) | **14.76** | 18.57 | 44.54 | 27.99 | 37.82 | 15.43 | 37.93 | 44.07 | 62.26 | 33.71 |
| UpIT(8E,A2) | 14.63 | **21.92** | 47.04 | 29.25 | 39.84 | 14.49 | 36.86 | 46.78 | 60.85 | **34.63** |
| *Results of 200K training samples* | | | | | | | | | | |
| SFT | 9.15 | 16.15 | 44.90 | 29.03 | 34.73 | **16.04** | **40.59** | 44.07 | 60.67 | 32.81 |
| Upcycle$_{PT}$(4E,A2) | 19.51 | 24.26 | 50.84 | 29.08 | 37.95 | 13.82 | 34.72 | 46.44 | 59.79 | 35.16 |
| Upcycle$_{PT}$(8E,A2) | 18.29 | 22.90 | 47.47 | **30.08** | 37.82 | 13.91 | 35.23 | 50.51 | 67.72 | 35.99 |
| UpIT(4E,A2) | 20.63 | 25.55 | 50.92 | 29.08 | 38.72 | 15.37 | 35.84 | 49.49 | 61.20 | 36.31 |
| UpIT(8E,A2) | **21.34** | **28.73** | 47.47 | 29.08 | 39.84 | 15.43 | 36.19 | 50.51 | 66.49 | **37.23** |
| *Results of 500K training samples* | | | | | | | | | | |
| SFT | 26.22 | 29.19 | 38.01 | 26.46 | 33.93 | 8.42 | 18.61 | 42.37 | 58.55 | 31.31 |
| Upcycle$_{PT}$(4E,A2) | 31.71 | 35.10 | 43.40 | **30.23** | 37.93 | 13.74 | 34.72 | 45.08 | 58.73 | 36.74 |
| Upcycle$_{PT}$(8E,A2) | 26.22 | 30.62 | 46.76 | 28.95 | 39.84 | 13.57 | 33.86 | 49.62 | 66.49 | 37.33 |
| UpIT(4E,A2) | 31.34 | 33.81 | 48.97 | 29.53 | 40.84 | **14.71** | 36.99 | 47.80 | 65.96 | 38.88 |
| UpIT(8E,A2) | **32.19** | **35.64** | 49.15 | 30.23 | 40.38 | 14.57 | 37.93 | 49.10 | 68.43 | **39.74** |

models during expert preparation, resulting in greater diversity among the expert models and thus a faster rate of performance growth.

Table 5: Detailed results of different numbers of experts and activated parameters, where (`xE`,`Ay`) indicates that `y` out of `x` experts are activated. LoRA-based UpIT(`16E`,`A2`) is expanded from LoRA-based UpIT(`8E`,`A2`). Bold text and underlined text denote the best and second-best results in each group.

| | HumanE. | GSM8K | HellaS. | BBH | MMLU | NQ | TriviaQA | ARC-c | ARC-e | Avg. |
|---|---|---|---|---|---|---|---|---|---|---|
| *LoRA-based upcycling models* | | | | | | | | | | |
| LoRAMoE$_{PT}$(8E,A1) | 29.27 | 41.32 | 48.71 | 34.35 | 38.17 | 16.68 | 41.11 | 55.25 | 72.49 | 41.93 |
| LoRAMoE$_{PT}$(8E,A2) | 34.15 | 47.61 | 60.89 | 37.40 | 46.61 | 17.62 | 46.33 | 60.68 | 74.60 | 47.32 |
| LoRAMoE$_{PT}$(8E,A4) | 39.63 | 48.82 | 63.88 | 37.26 | 49.25 | 20.39 | 50.10 | 65.42 | 80.42 | 50.57 |
| LoRAMoE$_{PT}$(8E,A6) | 41.46 | 50.95 | 66.70 | 37.46 | **50.65** | 22.33 | 50.80 | 64.07 | **80.95** | **51.71** |
| LoRAMoE$_{PT}$(16E,A1) | 27.44 | 36.09 | 43.58 | 33.05 | 39.58 | 15.57 | 36.79 | 51.96 | 65.61 | 38.85 |
| LoRAMoE$_{PT}$(16E,A2) | 40.24 | 46.55 | **65.89** | 36.39 | 48.53 | 19.36 | 46.19 | 61.69 | 76.01 | 48.98 |
| LoRAMoE$_{PT}$(16E,A4) | **42.68** | 49.96 | 58.65 | 37.59 | 48.64 | 19.39 | 50.94 | 65.08 | 79.19 | 50.24 |
| LoRAMoE$_{PT}$(16E,A6) | 42.07 | **51.05** | 64.19 | **37.77** | 49.84 | 22.33 | **51.60** | 62.71 | 80.42 | 51.33 |
| UpIT(8E,A1) | 20.73 | 43.59 | 63.51 | 40.01 | 49.13 | 20.86 | 48.10 | 62.03 | 80.78 | 47.64 |
| UpIT(8E,A2) | 35.37 | 49.51 | 66.00 | 40.27 | 50.31 | 24.52 | 55.27 | 65.08 | 83.60 | 52.21 |
| UpIT(8E,A4) | 40.12 | 49.05 | 65.82 | 39.98 | 51.15 | **25.96** | 57.50 | 67.80 | 84.66 | 53.56 |
| UpIT(8E,A6) | 43.62 | 49.13 | 65.81 | 39.53 | 50.96 | 25.48 | 55.57 | 67.89 | 85.19 | 53.69 |
| UpIT(16E,A1) | 21.95 | 40.71 | 61.47 | 40.19 | 49.39 | 24.99 | 57.62 | 64.07 | 81.31 | 49.08 |
| UpIT(16E,A2) | 40.62 | 48.37 | **66.62** | 39.43 | 50.70 | 25.62 | 56.61 | 67.46 | 84.66 | 53.34 |
| UpIT(16E,A4) | 42.53 | 50.49 | 65.85 | **41.08** | 51.13 | 25.29 | 56.80 | 66.78 | 84.83 | 53.86 |
| UpIT(16E,A6) | **43.62** | **51.25** | 66.14 | 40.74 | **51.33** | 25.48 | 56.95 | 67.12 | **85.19** | **54.20** |
| *FFN-based upcycling models* | | | | | | | | | | |
| Upcycle$_{PT}$(4E,A1) | 13.20 | 26.73 | 42.73 | 29.28 | 38.72 | 13.02 | 32.08 | 49.15 | 62.26 | 34.13 |
| Upcycle$_{PT}$(4E,A2) | 31.71 | **35.10** | 43.40 | **30.23** | 37.93 | 13.74 | 34.72 | 45.08 | 58.73 | 36.74 |
| Upcycle$_{PT}$(4E,A4) | **32.93** | 33.89 | **51.95** | 29.62 | 39.16 | **13.91** | 34.16 | 49.83 | **67.55** | **39.22** |
| Upcycle$_{PT}$(8E,A1) | 11.68 | 23.28 | 41.68 | 28.11 | 37.93 | 13.49 | 30.73 | 48.92 | 61.39 | 33.02 |
| Upcycle$_{PT}$(8E,A2) | 26.22 | 34.04 | 51.57 | 28.95 | 39.24 | 13.57 | 33.86 | 53.22 | 66.49 | 38.64 |
| Upcycle$_{PT}$(8E,A4) | 25.61 | 34.27 | 50.59 | 30.12 | **40.63** | 13.68 | 34.98 | 53.22 | 67.02 | 38.90 |
| UpIT(4E,A1) | 19.51 | 28.73 | 46.81 | 29.56 | 37.69 | 14.40 | 34.56 | 48.81 | 66.31 | 36.26 |
| UpIT(4E,A2) | 31.34 | 33.81 | 48.97 | 29.53 | **40.84** | 14.71 | 36.99 | 47.80 | 65.96 | 38.88 |
| UpIT(4E,A4) | 33.43 | **37.62** | 48.02 | 29.25 | 40.76 | **15.28** | **40.19** | 47.46 | **69.19** | 40.13 |
| UpIT(8E,A1) | 20.12 | 30.40 | 42.76 | 29.44 | 37.98 | 13.99 | 35.71 | 50.17 | 66.31 | 36.32 |
| UpIT(8E,A2) | 32.19 | 35.64 | 49.15 | **30.23** | 40.38 | 14.57 | 37.93 | 49.10 | 68.43 | 39.74 |
| UpIT(8E,A4) | **34.56** | 36.73 | **50.27** | 29.17 | 40.76 | 14.79 | 37.49 | **51.26** | 68.25 | **40.36** |

## A.6 DETAIL RESULTS OF FURTHER ANALYSIS

### A.6.1 DETAIL RESULTS OF DIFFERENT STRATEGIES DURING EXPERT PREPARATION

Table 8 shows the detailed results of different checkpoint selection approaches during expert preparation. We find that utilizing the latter half of the checkpoints as expert models yields stronger performance, particularly in mathematical reasoning and code generation capabilities. We believe this is due to the continuous improvement of mathematical reasoning and code generation as the volume of data or training increases. This observation aligns with the conclusions in (Li et al., 2024a), which indicate that selecting the latter half of the checkpoints can enhance mathematical reasoning and code generation abilities.

### A.6.2 DETAIL RESULTS OF DIFFERENT STRATEGIES DURING EXPERT EXPANSION

Table 9 shows the detailed results of different expert expanding strategies during expert expansion. We are pleasantly surprised to find that our genetic algorithm-based method demonstrates highly competitive performance compared to using all expert models, indicating that the merged expert models indeed possess sufficient diversity. In contrast, the approach of randomly selecting expert models to construct new experts results in a decline in performance, which can be attributed to the insufficient diversity among the randomly chosen expert models.

### A.6.3 DETAIL RESULTS OF DIFFERENT STRATEGIES DURING ROUTER INITIALIZATION

Table 10 shows the detailed results of different expert-specific data selection approaches. We find that our proposed PPL-based data selection method achieves the best performance. Interestingly, the method of randomly selecting expert-specific data has a detrimental effect. We believe that randomly selected data leads to a reduction in the diversity of the experts, resulting in poorer performance.

Table 6: Detailed results of performance upper bound with LoRA-based upcycling. All models are under (8E, A2) settings and (1) represents totally 1 training epoch. Bold text and underlined text denote the best and second-best results in each group.

| | HumanE. | GSM8K | HellaS. | BBH | MMLU | NQ | TriviaQA | ARC-c | ARC-e | Avg. |
|---|---|---|---|---|---|---|---|---|---|---|
| *dense LLMs* | | | | | | | | | | |
| LoRA(1) | 15.85 | 40.26 | 50.54 | **40.40** | 49.12 | 12.60 | **60.50** | **64.07** | **82.19** | 46.17 |
| LoRA(2) | 22.56 | 43.90 | 58.03 | 39.25 | 45.61 | 13.30 | 59.15 | 63.73 | 81.31 | 47.43 |
| LoRA(3) | 17.68 | 46.10 | 58.79 | 38.88 | 47.97 | 14.68 | 57.84 | **64.07** | 81.66 | 47.52 |
| LoRA(4) | 22.56 | 45.72 | **65.36** | 37.14 | **49.33** | 14.99 | 50.15 | 61.36 | 81.13 | 47.53 |
| LoRA(5) | 21.62 | 47.71 | 63.29 | 38.88 | 48.15 | 13.30 | 53.64 | 61.36 | 82.01 | **47.77** |
| LoRA(6) | 23.17 | 47.61 | 64.89 | 40.46 | 47.49 | 15.07 | 50.80 | 60.68 | 74.60 | 47.20 |
| LoRA(7) | **26.22** | 45.11 | 50.84 | 38.80 | 48.03 | **15.43** | 54.72 | **64.07** | **82.19** | 47.27 |
| LoRA(8) | 25.73 | **47.96** | 65.30 | 37.26 | 47.35 | 14.07 | 46.19 | 61.36 | 82.01 | 47.47 |
| *LoRA-based upcycling models* | | | | | | | | | | |
| LoRAMoE$_{PT}$(1) | 25.61 | 43.90 | 52.49 | **39.21** | **51.17** | **24.71** | **56.29** | 61.36 | 77.95 | 48.08 |
| LoRAMoE$_{PT}$(2) | 31.71 | 48.60 | 55.33 | 38.69 | 47.92 | 22.83 | 51.18 | 62.03 | 76.37 | 48.30 |
| LoRAMoE$_{PT}$(3) | 35.56 | **49.66** | 54.78 | 38.25 | 50.60 | 20.80 | 50.15 | 60.36 | 77.10 | **48.58** |
| LoRAMoE$_{PT}$(4) | 34.15 | 47.61 | 60.89 | 37.40 | 46.61 | 17.62 | 46.33 | 60.68 | 74.60 | 47.32 |
| LoRAMoE$_{PT}$(5) | 35.98 | 48.82 | 63.10 | 35.84 | 48.15 | 18.20 | 44.52 | 62.71 | 78.48 | 48.42 |
| LoRAMoE$_{PT}$(6) | 35.98 | 46.47 | 63.20 | 37.60 | 47.40 | 16.26 | 42.45 | 62.03 | 76.37 | 47.53 |
| LoRAMoE$_{PT}$(7) | 33.54 | 48.82 | 62.71 | 36.99 | 47.17 | 16.48 | 41.60 | **65.08** | **78.66** | 47.89 |
| LoRAMoE$_{PT}$(8) | **37.80** | 46.47 | **63.71** | 37.29 | 47.88 | 15.96 | 38.84 | **65.08** | 76.37 | 47.71 |
| *LoRA-based UpIT with 2 epochs expert preparation* | | | | | | | | | | |
| UpIT(2,1) | 32.93 | 49.13 | 64.89 | 39.73 | 49.39 | 25.48 | 54.05 | 64.07 | 80.76 | 51.16 |
| UpIT(2,2) | 35.37 | 49.51 | 66.00 | **40.27** | 50.31 | 24.52 | 55.27 | 65.08 | 83.60 | 52.21 |
| UpIT(2,3) | 36.48 | 50.29 | **67.19** | 38.80 | 49.88 | 25.12 | 54.72 | **68.81** | 82.19 | 52.61 |
| UpIT(2,4) | 37.69 | 50.49 | 65.85 | 41.08 | 50.96 | 25.69 | 56.95 | 67.13 | 84.91 | 53.42 |
| UpIT(2,5) | 39.12 | 50.95 | 66.70 | 40.27 | 51.13 | 25.48 | **58.54** | 68.03 | **85.19** | 53.93 |
| UpIT(2,6) | **41.46** | **51.93** | 66.14 | 41.08 | **51.33** | **26.59** | 57.61 | **68.81** | 84.66 | **54.40** |
| *LoRA-based UpIT with 4 epochs expert preparation* | | | | | | | | | | |
| UpIT(4,1) | 34.39 | 48.22 | 70.10 | 39.43 | 51.45 | 25.58 | 53.62 | 63.43 | 78.48 | 51.63 |
| UpIT(4,2) | 37.82 | 48.98 | 70.24 | 39.04 | 51.52 | 25.72 | 54.72 | 64.56 | 80.32 | 52.55 |
| UpIT(4,3) | 38.62 | 49.58 | 70.63 | 39.32 | 51.10 | 25.37 | 57.29 | 66.79 | 83.42 | 53.57 |
| UpIT(4,4) | **40.12** | **50.82** | **70.98** | **40.27** | **51.93** | **26.19** | **57.37** | **67.74** | **84.69** | **54.46** |

Table 7: Detailed results of performance upper bound with FFN-based upcycling. All models are under `(4E,A2)` settings and `(1)` represents totally 1 training epoch. Bold text and underlined text denote the best and second-best results in each group.

| | HumanE. | GSM8K | HellaS. | BBH | MMLU | NQ | TriviaQA | ARC-c | ARC-e | Avg. |
|---|---|---|---|---|---|---|---|---|---|---|
| *dense LLMs* | | | | | | | | | | |
| SFT(1) | 17.07 | 24.94 | 30.10 | **28.29** | 29.77 | 8.01 | 20.48 | 40.68 | 46.56 | 27.32 |
| SFT(2) | 19.51 | 27.98 | 27.59 | 26.69 | 32.41 | 8.53 | 20.35 | 38.98 | 55.91 | 28.66 |
| SFT(3) | 20.73 | 31.77 | 40.28 | 26.23 | **34.54** | **9.22** | 20.81 | 42.03 | **58.73** | 31.59 |
| SFT(4) | **26.22** | 29.19 | 38.01 | 26.46 | 33.93 | 8.42 | 18.61 | 42.37 | 58.55 | 31.31 |
| SFT(5) | 24.39 | 25.32 | 39.29 | 27.28 | 32.99 | 8.53 | **21.15** | 43.39 | 53.44 | 30.64 |
| SFT(6) | 22.56 | **32.07** | **41.67** | 27.82 | 34.35 | 9.00 | 17.20 | **44.75** | 55.03 | **31.61** |
| SFT(7) | 21.34 | 30.10 | 37.60 | 27.14 | 33.25 | 8.59 | 19.55 | 42.71 | **58.73** | 31.00 |
| SFT(8) | 22.56 | 29.42 | 41.16 | 26.46 | 32.26 | 8.48 | 18.18 | 34.58 | 51.32 | 29.38 |
| *FFN-based upcycling models* | | | | | | | | | | |
| Upcycle$_{PT}$(1) | 18.29 | 31.54 | 42.58 | 29.42 | 35.54 | 14.71 | **35.18** | 49.49 | 65.96 | 35.86 |
| Upcycle$_{PT}$(2) | 20.73 | **37.23** | **51.79** | 28.29 | **40.33** | 14.24 | 34.12 | 47.80 | 64.02 | 37.62 |
| Upcycle$_{PT}$(3) | 26.22 | 34.80 | 45.66 | 30.12 | 38.52 | **14.79** | 34.59 | 49.49 | **66.31** | 37.83 |
| Upcycle$_{PT}$(4) | **31.71** | 35.10 | 43.40 | 30.23 | 37.93 | 13.74 | 34.72 | 45.08 | 58.73 | 36.74 |
| Upcycle$_{PT}$(5) | 28.05 | 35.03 | 48.59 | 30.22 | 37.08 | 13.91 | 34.57 | **50.85** | 65.26 | **38.17** |
| Upcycle$_{PT}$(6) | 28.05 | 35.41 | 43.40 | 29.93 | 39.27 | 13.93 | 33.07 | 50.51 | 61.39 | 37.22 |
| Upcycle$_{PT}$(7) | 29.27 | 35.03 | 46.26 | 30.30 | 38.18 | 14.24 | 33.04 | 49.49 | 62.26 | 37.56 |
| Upcycle$_{PT}$(8) | 30.62 | 35.71 | 44.19 | **30.71** | 35.99 | 10.75 | 33.02 | 47.80 | 62.43 | 36.80 |
| *FFN-based UpIT with 2 epochs expert preparation* | | | | | | | | | | |
| UpIT(2,1) | 17.07 | 29.11 | 45.77 | **31.03** | 36.07 | 15.24 | **38.34** | 43.05 | 68.43 | 36.01 |
| UpIT(2,2) | 31.34 | 33.81 | 48.97 | 29.53 | 40.84 | 14.71 | 36.99 | 47.80 | 65.96 | 38.88 |
| UpIT(2,3) | 33.78 | 35.03 | 51.43 | 29.75 | **41.22** | 14.88 | 35.09 | 55.59 | 64.02 | 40.09 |
| UpIT(2,4) | 35.69 | 37.62 | 51.06 | 28.06 | 40.88 | 14.16 | 34.39 | **56.08** | 67.90 | 40.65 |
| UpIT(2,5) | 36.75 | **38.61** | 50.59 | 28.62 | 38.52 | 15.24 | 37.12 | 55.59 | **68.96** | 41.11 |
| UpIT(2,6) | **39.12** | 37.62 | **51.95** | 30.12 | 39.60 | **15.96** | 36.18 | 55.59 | 66.67 | **41.42** |
| *FFN-based UpIT with 4 epochs expert preparation* | | | | | | | | | | |
| UpIT(4,1) | 17.07 | 27.37 | 43.90 | 29.84 | 38.92 | **15.40** | **38.30** | 50.17 | 64.55 | 36.17 |
| UpIT(4,2) | 30.62 | 31.08 | 48.74 | 30.62 | 39.69 | 13.71 | 37.38 | 49.83 | **70.90** | 39.17 |
| UpIT(4,3) | 34.69 | 37.23 | **51.73** | **30.84** | 39.64 | 13.38 | 35.18 | 53.22 | 70.19 | 40.68 |
| UpIT(4,4) | **38.92** | **40.26** | 50.56 | 30.05 | **39.81** | 14.79 | 36.18 | **53.61** | 69.49 | **41.52** |

Table 8: Detailed results of different checkpoint selection strategies during expert preparation. `Front.Half` represents selecting the first half of checkpoints, `Uniform` represents uniformly selecting checkpoints and `Back.Half` represents selecting the back half ones which is used in our paper. All models are LoRA-based models under (8E,A2)

| | HumanE. | GSM8K | HellaS. | BBH | MMLU | NQ | Tri.QA | ARC-c | ARC-e | Avg. |
|---|---|---|---|---|---|---|---|---|---|---|
| UpIT(Front.Half) | 21.34 | 44.88 | 64.48 | 40.55 | **51.36** | **27.56** | **59.36** | 68.47 | 84.30 | 51.37 |
| UpIT(Uniform) | 27.44 | 43.90 | 65.39 | **40.73** | 50.93 | 25.90 | 58.31 | 68.14 | **84.66** | 51.71 |
| UpIT(Back.Half) | **35.37** | **49.51** | **66.00** | 40.27 | 50.31 | 24.52 | 55.27 | 65.08 | 83.60 | **52.21** |

Table 9: Detailed results of different expanding strategies during expert expansion. `w/o EE` represents directly using checkpoints for expert preparation without expert expansion. `Random` represents randomly selecting two experts to merge a new one during expert expansion and `Genetic` represents the selection approach shown in Algorithm 3. All models are LoRA-based models under (16E,A2)

| | HumanE. | GSM8K | HellaS. | BBH | MMLU | NQ | Tri.QA | ARC-c | ARC-e | Avg. |
|---|---|---|---|---|---|---|---|---|---|---|
| UpIT(w/o EE) | **40.62** | **48.82** | 65.58 | **40.60** | **51.59** | 25.24 | **57.00** | 67.12 | 83.25 | 53.31 |
| UpIT(Random) | 38.72 | 44.71 | 66.14 | 40.27 | 50.96 | 25.48 | 54.72 | 67.13 | 83.60 | 52.41 |
| UpIT(Genetic) | **40.62** | 48.37 | **66.62** | 39.43 | 50.70 | **25.62** | 56.61 | **67.46** | **84.66** | **53.34** |

Table 10: Detailed results of different data selection strategies during router initialization. `w/o Init` represents training models without our proposed router initialization. `Random` represents randomly construct the expert-specific data and `Skilled` represents our PPL-based data selection method as shown in Algorithm 3. All models are LoRA-based models under (8E,A2).

| | HumanE. | GSM8K | HellaS. | BBH | MMLU | NQ | Tri.QA | ARC-c | ARC-e | Avg. |
|---|---|---|---|---|---|---|---|---|---|---|
| UpIT(w/o Init) | 32.32 | **49.81** | **66.86** | 39.15 | 48.88 | 21.08 | 49.67 | 62.71 | 79.19 | 49.96 |
| UpIT(Random) | 28.05 | 46.70 | 64.27 | 38.67 | 49.63 | 21.11 | 48.67 | 64.41 | 82.19 | 49.30 |
| UpIT(Skilled) | **35.37** | 49.51 | 66.00 | **40.27** | **50.31** | **24.52** | **55.27** | **65.08** | **83.60** | **52.21** |

