# OpenReview forum: "Upcycling Instruction Tuning from Dense to Mixture-of-Experts via Parameter Merging"
_ICLR.cc/2025/Conference — Submitted to ICLR 2025_

### Official Review · Reviewer_WPEf · 2024-11-03

**Soundness:** 3
**Presentation:** 3
**Contribution:** 3
**Rating:** 5
**Confidence:** 4

**Summary:**

This paper introduces "Upcycling Instruction Tuning" (UpIT), a novel and data-efficient method for converting dense pre-trained language models into a Mixture-of-Experts (MoE) model. UpIT leverages intermediate checkpoints from the dense model’s instruction-tuning phase to create specialized "experts" and employs a genetic algorithm for expert expansion to maintain diversity and flexibility in expert numbers. Additionally, it uses a router initialization step to optimize the model’s token routing, allowing each expert to excel in designated tasks. Compared to traditional methods, UpIT significantly reduces data requirements while improving performance, scalability, and stability in MoE models across various natural language processing benchmarks. The paper’s extensive experiments validate UpIT's effectiveness, particularly in low-data scenarios and when scaling expert counts, highlighting the importance of expert diversity for efficient model upcycling.

**Strengths:**

1. The paper is well-organized and clearly written, which is very easy to follow.
2. The proposed method is reasonable: by specializing a few experts and further merging and mutating into new ones.
3. The authors provided comprehensive experiments.

**Weaknesses:**

**Ablation Study on Hyperparameters m and n in Algorithm 2**: The lack of an ablation study comparing values of m and n (step 1 in Algorithm 2) leaves an important gap in understanding how to control expert diversity and scalability in expert merging. This comparison is likely critical, as it would shed light on the trade-offs involved in selecting these hyperparameters, such as the extent of expert specialization versus computational efficiency. Providing guidance on optimal values for different model sizes and data conditions would greatly assist practitioners in fine-tuning their model configurations.

**Questions:**

1. **Specialized Expert Preparation and Data Distributions**: How can we ensure that the preparation of specialized experts (step 2 in Algorithm 1) effectively captures diverse expertise, especially when data distribution is varied? While Figure 3 only considers different scales of training data, it would be insightful to evaluate how experts perform with data that varies not only in size but also in domain or content distribution. Would experts trained on domain-specific or contextually diverse subsets offer improved performance or encounter challenges in maintaining consistent expertise?

2. **Forgetting Issues in Expert Specialization**: When specializing the dense model into distinct experts using diverse datasets, some experts may experience different levels of forgetting, particularly for knowledge outside their specialized domain. Would it be beneficial to retain a copy of the original dense model throughout the expert expansion process to mitigate potential knowledge loss? Retaining this copy might help rebalance the model’s general knowledge base, especially if certain experts regress on commonly shared tasks.

---

> ### Author Response · Authors · 2024-11-20
>
> We are encouraged that you find UpIT is **reasonable and easy-to-follow**. Thanks for the valuable comments that could help us improve the work. We will respond to the questions, and *we would appreciate it if you could kindly raise the scores when the concerns are addressed.*
>
> > **W1: Lack of ablation study on m and n in Algorithm 2.**
>
> Thanks for your insightful comments. In Figure 4, we explore ablation studies with different values of m and different activation experts, and the detailed results are provided in Table 5. It can be found that UpIT achieves excellent performance under different total and activated experts. Regarding the ablation study on n, we are trying our best to supplement the experiment and will provide it soon.
>
> > **Q1.1: How can we ensure that the preparation of specialized experts effectively captures diverse expertise.**
>
> As shown in Figure 2 and Table 6,7, The intermediate checkpoints in the instruction fine-tuning stage are naturally different and do not require special operations or attention. It introduces our first contribution of using intermediate checkpoints for expert preparation and corresponding expert expansion strategies. On this basis, we propose a router initialization stage with expert-specific data selection, utilizing and amplifying the differences among experts in the initialization phase. The better performance in Table 1 and data-efficient characteristics in Figure 3 confirm the above statement.
>
> > **Q1.2: Would experts trained on domain-specific or contextually diverse subsets offer improved performance or encounter challenges in maintaining consistent expertise?**
>
> Specialized upcycling trains a specific expert for each domain, which lacks flexibility as it is difficult to divide a mixed general dataset into separate domains. In constant, UpIT considers more universal scenarios, in which data comes from a mixed set and cannot be effectively divided. Traditional specialized upcycling methods are no longer applicable, while UpIT achieves efficient and flexible upcycling by seeking natural differentiation between intermediate checkpoints.
>
> > **Q2: Forgetting issues in expert specialization.**
>
> Thanks for your suggestion. We conduct experiments by incorporating the pre-trained model as an expert, in both configurations: one without and one with an expert expansion stage.
>
> | Method | HumanE. | GSM8K   | HellaS. | BBH    | MMLU   | NQ     | Tri.QA | ARC_c  | ARC_e  | Avg |
> |--------|---------|---------|---------|--------|--------|--------|--------|---------|---------|---------|
> | Llama 2-7B | 14.63 | 13.95 | 26.58 | 34.73 | 39.84 | 10.06 | 62.06 | 37.29 | 50.26 | 32.16 |
> | UpIT (8E,A2) | **35.37** | **49.51** | **66.00** | 40.27 | 50.31 | **24.52** | 55.27 | **65.08** | **83.60** | 52.21 |
> | UpIT (8E,A2) w/ pre-trained | 34.15 | 49.33 | 65.36 | **40.78** | **50.96** | 23.75 | **58.73** | 64.71 | 82.26 | **52.23** |
> | UpIT (16E,A2) | **40.62** | **48.37** | 66.62 | **39.43** | 50.70 | **25.62** | 56.61 | **67.46** | **84.66** | **53.34** |
> | UpIT (16E,A2) w/ pre-trained | 40.13 | 47.67 | **66.76** | 38.20 | **51.32** | 25.47 | **58.11** | 66.49 | 83.82 | 53.11 |
>
> Compared to Llama 2-7B, the fine-tuned models exhibit a performance decline on the TriviaQA benchmark. By incorporating the pre-trained model into the UpIT pipeline (note that only intermediate checkpoints are used in the paper), there is a considerable recovery in performance on TriviaQA. However, the performance of other benchmarks are declined, especially on HumanEval and GSM8K,  as the pre-trained model is not satisfactory on these tasks. It raises the issue of the trade-off between downstream performance and pre-trained knowledge, and we believe is a topic worthy of in-depth exploration in future endeavors.
>
> ---
>
> Thank you once again for your valuable feedback we are committed to refining our work based on your insights.
>
> If there are any remaining concerns or questions that we have not yet addressed, please do not hesitate to let us know. We are more than willing to provide any additional materials or clarifications that may be necessary.

---

> > ### Comment · Reviewer_WPEf · 2024-11-26
> > **Thanks for the response**
> >
> > I thank the authors' response.
> >
> > > **W1**
> >
> > I don't think Fig. 4 answered my question. I think Fig. 4 is about different numbers of activated experts, whereas I was asking for different configurations of $m, n$, where $m$ is the number of pretrained checkpoints and $n$ is after duplication. If $m$ equals to the number of activated experts, the authors should explicitly point that out in Fig. 4 caption.
> >
> > I also hope the authors could discuss potential limitations (Q1.2, Q2) in the paper, as this could be helpful for readers when considering the practical usage of this method.

---

> > > ### Author Response · Authors · 2024-11-26
> > >
> > > Thank you for your suggestions. We will provide the ablation study on m soon.
> > >
> > > Regarding the concerns mentioned in Q1.2 and Q2, we will include relevant content in the next version.

---

> > > ### Author Response · Authors · 2024-11-29
> > >
> > > We conducted an ablation experiment on m, and the results are as follows:
> > >
> > > | Method | HumanE. | GSM8K   | HellaS. | BBH    | MMLU   | NQ     | Tri.QA | ARC_c  | ARC_e  | Avg |
> > > |--------|---------|---------|---------|--------|--------|--------|--------|---------|---------|---------|
> > > | UpIT (4E,A2) | 34.21 | 47.20 | 63.92 | 38.79 | 49.15 | 24.07 | 46.38 | 63.72 | 81.93 | 49.93 |
> > > | UpIT (8E,A2) | 35.37 | 49.51 | 66.00 | 40.27 | 50.31 | 24.52 | 55.27 | 65.08 | 83.60 | 52.21 |
> > > | UpIT (12E,A2) | 37.51 | **50.33** | **66.18** | 39.90 | 50.48 | 25.09 | 55.61 | 65.77 | **83.91** | 52.75 |
> > > | UpIT (16E,A2) | **40.62** | 48.82 | 65.58 | **40.60** | **51.59** | **25.24** | **57.00** | **67.12** | 83.25 | **53.31** |
> > >
> > > We find that as the number of experts increases, the performance also gradually improves.
> > >
> > > ---
> > >
> > > If we have addressed your concerns, we would be delighted if you could consider raising the score.

---

> > > > ### Comment · Reviewer_WPEf · 2024-11-30
> > > > **I still have questions**
> > > >
> > > > Sorry I am a bit confused: are 4E/8E/12E/16E indicating $m$ or $n$? I was thinking of a kind of "trade-off" between $m$ and $n$ (i.e. diversity and scalability). This was explicitly mentioned in my original review "Weakness" section. Higher $m$ means higher diversity, and higher $n$ means higher scalability.
> > > >
> > > > Moreover, from your table, does that mean the higher $m$ or $n$ is always the better? It cannot tell me how to properly choose $m$ or $n$ in my experiment.

---

> > > > > ### Author Response · Authors · 2024-11-30
> > > > >
> > > > > Thank you for your valuable response.
> > > > >
> > > > > Our first table represents the use of 8 intermediate checkpoints, followed by the expansion of existing experts, which is to fix $m$ and scale $n$, resulting in the expansion of 2 (10E), 4 (12E), 6 (14E), and 8 (16E) new experts respectively. We observe that as the number of $n$ increases, the performance continues to grow.
> > > > >
> > > > > For the second table, we used 4 to 16 intermediate checkpoints without expanding the experts, that is, scaling $m$ directly. We find that the performance also continues to increase.
> > > > >
> > > > > Regarding your question about the trade-off between $m$ and $n$, we believe it depends on the actual needs because comparing the two experiments, we find that under the same number of experts, the performance difference between using expert expansion and directly using intermediate checkpoints is minimal. Therefore, if there are enough intermediate checkpoints, expert expansion can be omitted, and if the number of intermediate checkpoints is insufficient, expert expansion can be appropriately carried out to meet specific needs.
> > > > >
> > > > > In our experiments, as $m$ or $n$ increases, the performance continues to rise, but due to the limitations of computational resources, we cannot further increase $m$ or $n$ to verify when the performance begins to decline.

---

> > > > > > ### Author Response · Authors · 2024-12-03
> > > > > >
> > > > > > Dear reviewer,
> > > > > >
> > > > > > As the discussion period is nearing its conclusion, we earnestly hope to receive your feedback on our paper. We are keenly aware of the valuable insights you can provide, and we are hopeful that any outstanding concerns or questions will be communicated to us promptly. Your comments are crucial for the finalization of our work, and we are prepared to offer any further materials, experimental details, or clarifications that you may require.
> > > > > >
> > > > > > We are deeply appreciative of your expertise and are confident that your feedback will significantly contribute to the enhancement of our research. We look forward to your response before the discussion period ends.
> > > > > >
> > > > > > Best regards,
> > > > > >
> > > > > > The authors,

---

> ### Author Response · Authors · 2024-11-26
>
> > **W1: Lack of ablation study on m and n in Algorithm 2.**
>
> We have completed the ablation study for n, as shown in the table below.
>
> | Method | HumanE. | GSM8K   | HellaS. | BBH    | MMLU   | NQ     | Tri.QA | ARC_c  | ARC_e  | Avg |
> |--------|---------|---------|---------|--------|--------|--------|--------|---------|---------|---------|
> | UpIT (8E,A2) | 35.37 | **49.51** | 66.00 | **40.27** | 50.31 | 24.52 | 55.27 | 65.08 | 83.60 | 52.21 |
> | UpIT (10E,A2) | 36.32 | 48.76 | 66.08 | 39.82 | 50.43 | 24.77 | 54.85 | 66.10 | 83.91 | 52.34 |
> | UpIT (12E,A2) | 37.82 | 49.03 | 65.67 | 40.03 | 50.75 | 25.13 | 55.93 | 66.38 | 84.10 | 52.76 |
> | UpIT (14E,A2) | 39.40 | 48.41 | 66.39 | 39.38 | **50.91** | 25.48 | 56.38 | 67.04 | 84.10 | 53.05 |
> | UpIT (16E,A2) | **40.62** | 48.37 | **66.62** | 39.43 | 50.70 | **25.62** | **56.61** | **67.46** | **84.66** | **53.34** |
>
> We employ the expert expansion phase to incorporate 2, 4, and 6 new experts, respectively. The results demonstrate that with the progressive integration of additional experts, there is a corresponding enhancement in the average performance.
>
> ---
>
> Thank you once again for your valuable feedback we are committed to refining our work based on your insights.
>
> If we have addressed your concerns, we would be delighted if you could consider raising the score.

---

### Official Review · Reviewer_emDR · 2024-11-03

**Soundness:** 2
**Presentation:** 3
**Contribution:** 3
**Rating:** 5
**Confidence:** 3

**Summary:**

Based on the pilot experiment, which showed that different checkpoints during instruction tuning are inherently suitable for creating specialized experts, they proposed a method that leverages intermediate dense checkpoints for model upcycling. Additionally, to address the issue of routers being randomly initialized after upcycling—leading to token misallocation in the early post-training stage—they introduced a data selection strategy to curate expert-specific datasets tailored to each expert model and pre-optimize additional routing vectors to enhance differentiation among experts. They demonstrated the effectiveness of their methods on various benchmarks, outperforming other upcycling baselines in LoRA-based and FFN-based upcycling settings.

**Strengths:**

1. The writing is clear, and they provide sufficient supplementary information to help readers easily understand the algorithm.
2. Their findings from the pilot experiment somewhat support their motivation for using the intermediate checkpoints.
3. They demonstrated the effectiveness of their methods on several benchmarks.

**Weaknesses:**

1. Looking at Figure 2, it appears that, across intervals, there is not much difference in performance for certain benchmarks. Therefore, the authors' claim that checkpoints saved at regular intervals can be considered specialized experts seems somewhat exaggerated.
2. It is unclear whether the same pattern would be observed with datasets other than the training dataset used in the paper. In other words, it needs to be verified if using checkpoints from different intervals for upcycling generalizes well across a variety of datasets.
3. In addition to explaining how they select checkpoints when the number of saved checkpoints exceeds the required number of experts, studying the impact of saving checkpoints at different intervals during dense model training could further enhance the robustness of the proposed method.

**Questions:**

1. Regarding the expert preparation process, how do we determine the appropriate interval for saving checkpoints? Is this solely based on heuristic methods?
2. In the expert expansion process, the two models with the greatest discrepancy are selected for merging. In this case, wouldn't this mostly result in the selection of checkpoints from the early and late stages of training?
3. How is the similarity between the two experts calculated?
4. During the training of the dense model, the training data was randomly sampled. What would happen if this approach was not used?

---

> ### Author Response · Authors · 2024-11-20
>
> Thank you very much for your recognition of our work. Thanks for the valuable comments that could help us improve the work. We will respond to the questions, and *we would appreciate it if you could kindly raise the scores when the concerns are addressed.*
>
> > **W1:  In Figure 2, there is not much difference in performance for certain benchmarks.**
>
> In this paper, we aim to emphasize the importance of expert discrepancy in upcycling and incorporate the idea into the entire design of UpIT. Figure 2 first highlights the different checkpoints during instruction tuning that are inherently suitable for constructing specialized experts, since different checkpoints excel at different benchmarks, which introduces our first contribution of using intermediate checkpoints for expert preparation and corresponding expert expansion strategies.  On this basis, we propose a router initialization stage with expert-specific data selection, utilizing and amplifying the differences among experts in the initialization phase. Overall, the initial differences among experts are the foundation of UpIT, and we fully utilize and amplify these differences to improve performance and data efficiency.
>
> Note that we present the results of multiple benchmarks in a single diagram, with a wide y-axis range, making it visually appear that the effects of different checkpoints are similar. However, in reality, they have significant differences (Table 6,7 provide more accurate numerical comparisons during an instruction tuning stage).
>
> > **W2: It is unclear whether the same pattern would be observed with other datasets.**
>
> Thanks for your suggestion. It is a common phenomenon that there are performance differences between different immediate checkpoints during instruction tuning. Here, we attach the performance across different checkpoints on the Alpaca dataset, and the results show similar conclusions to those presented in the paper.
>
> | Steps | HumanE. | GSM8K   | HellaS. | BBH    | MMLU   | NQ     | Tri.QA | ARC_c  | ARC_e  |
> |--------|---------|---------|---------|--------|--------|--------|--------|---------|---------|
> | 300   |    0.00     |    7.28     |    34.01     |    29.68    |   33.60     |    **21.80**    |    **51.60**    |     36.61    |    58.02     |
> | 600 | 12.80 | 5.61 | 25.30 | 31.38 | 35.79 | 21.41 | 51.54 | 42.71 | 66.14 |
> | 900 | 3.66 | 6.29 | 28.15 | **35.53** | 39.83 | 19.14 | 45.26 | **50.51** | **70.02** |
> | 1200 | 14.02 | 7.20 | **40.19** | 32.39 | 33.97 | 20.06 | 50.13 | 24.41 | 35.98 |
> | 1500 | 12.80 | **9.63** | 31.29 | 35.45 | 38.67 | 18.25 | 47.07 | 39.32 | 56.97 |
> | 1800 | 13.41 | 8.87 | 35.85 | 34.51 | **39.90** | 17.40 | 47.40 | 35.93 | 48.85 |
> | 2100 | **16.46** | 8.34 | 36.92 | 34.71 | 38.51 | 17.31 | 47.60 | 37.97 | 49.21 |
> | 2400 | 14.63 | 8.26 | 35.87 | 34.85 | 39.16 | 17.04 | 47.54 | 37.29 | 52.03 |
>
> We also observe similar patterns in the alpaca dataset, where there was a noticeable discrepancy at different checkpoints during the training process.
>
> > **W3: Studying the impact of saving checkpoints at different intervals during dense model training could further enhance the robustness of the proposed method.**
>
> The main viewpoint that the paper hopes to output is that using the natural discrepancy of intermediate checkpoints is able to achieve more efficient and effective upcycling. With this in mind, we mainly focus on the situation where the number of saved models is less than the expected number of experts - and then propose an expert expansion stage.
>
> For the sake of generalization and convenience, we do not emphasize the selection of checkpoints. When there are sufficient checkpoints available, various selection methods are applicable. In Table 2 (left), we illustrate three different selection methods: using the front half, back half, or average selection. Regardless of the selection method, there is a significant improvement compared to the vanilla upcycling. Therefore, we believe that as long as there is a discrepancy between experts during the expert preparation phase, the specific selection method becomes less important, and we keep the further checkpoint selection strategy for future work.
>
> | Method               | Average |
> |----------------------|---------|
> | $\text{LoRAMoE}_{\text{SFT}}$    | 49.99 |
> | UpIT (Front.Half)    | 51.37 |
> | UpIT (Uniform)      | 51.71 |
> | UpIT (Back.Half)    | 52.21 |

---

> ### Author Response · Authors · 2024-11-20
>
> > **Q1: How do we determine the appropriate interval for saving checkpoints?**
>
> Consistent with the previous response for W3, we do not pay special attention to the intervals for saving checkpoints, and adopt the naive approach of regularly saving checkpoints, which is also the most common practice in model training.
>
> > **Q2: Wouldn't this mostly result in the selection of checkpoints from the early and late stages of training?**
>
> In the parameter update process of model optimization, each parameter is not updated in a uniform and linear manner. Instead, it undergoes complex and dynamic changes, and the difference between the initial and final checkpoints may not be the largest. Furthermore, in order to create diverse experts during expert expansion, we place each newly created expert back into the pool of candidate experts, where it participates in the similarity calculation for the next round.
>
> > **Q3: How is the similarity between the two experts calculated?**
>
> The similarity is calculated based on the cosine similarity between the parameter matrices of two experts. Specifically, for a given LoRA expert's A and B matrices, we first compute the matrix product of A and B to obtain a parameter matrix E. Then, we calculate the cosine similarity between the parameter matrices E of different experts. The process for FFN experts is similar.
>
> > **Q4: What would happen if randomly sampled is not used?**
>
> In our experiment, we directly reuse the IDAE-500K dataset utilized in the previous baseline method [1], which is more convenient for experimental exploration on a data scale, as demonstrated in Figure 3 (with 50K, 100K, 200K, and 500K samples).
>
> [1] Wu et al. Parameter-Efficient Sparsity Crafting from Dense to Mixture-of-Experts for Instruction Tuning on General Tasks. EMNLP. 2024.
>
> ---
>
> Thank you once again for your valuable feedback we are committed to refining our work based on your insights.
>
> If there are any remaining concerns or questions that we have not yet addressed, please do not hesitate to let us know. We are more than willing to provide any additional materials or clarifications that may be necessary.

---

> > ### Author Response · Authors · 2024-11-29
> >
> > Dear reviewer,
> >
> > Thank you very much for the time and effort you have dedicated to reviewing our work.
> >
> > If our response satisfies you, we anticipate your reconsideration of the overall assessment of our paper. If there are any remaining concerns or questions that we have not yet addressed, please do not hesitate to let us know. We are more than willing to provide any additional materials, experimental details, or clarifications that may be necessary.
> >
> > We greatly value your insights and believe they will substantially enhance the quality of our work.
> >
> > Best regards,
> >
> > The authors,

---

> > > ### Author Response · Authors · 2024-12-03
> > >
> > > Dear reviewer,
> > >
> > > As the discussion period is nearing its conclusion, we earnestly hope to receive your feedback on our paper. We are keenly aware of the valuable insights you can provide, and we are hopeful that any outstanding concerns or questions will be communicated to us promptly. Your comments are crucial for the finalization of our work, and we are prepared to offer any further materials, experimental details, or clarifications that you may require.
> > >
> > > We are deeply appreciative of your expertise and are confident that your feedback will significantly contribute to the enhancement of our research. We look forward to your response before the discussion period ends.
> > >
> > > Best regards,
> > >
> > > The authors,

---

### Official Review · Reviewer_DUkz · 2024-11-04

**Soundness:** 2
**Presentation:** 2
**Contribution:** 2
**Rating:** 6
**Confidence:** 3

**Summary:**

This paper proposes UpIT, which aims to initialize MoE more data efficiently. The paper observes the optimal downstream tasks' performance usually happens in different training steps, so they use the checkpoints in different steps to initialize the experts. Furthermore, UpIT also uses model merging to create more experts and create expert-specific data to initialize routers.

**Strengths:**

The performance of UpIT is better than several baselines on LLaMA2, and the method scale with more data.

**Weaknesses:**

1. The paper claims the method is data-efficient. However, it is unclear how many samples each method uses during training in Table 1.

2. In Table 2, what would be the results of not using intermediate checkpoints to initialize MoE?

3. The proposed approach contains 3 main components (as shown in Table 2). What is the most important component that contributes to the final performance?

4. Also, UpIT is highly related to "Specialized Upcycling" as the training requires curated expert data. However, none of the baselines belong to specialized upcycling.

5. In line 258, "merging all the expert models" is confusing with the term model "merging" (parameter merging), and in line 258, it seems more like concatenation.

6. The contents in Figure 1 are very small.

7. In Sec 3.1, it would be great to explain the mechanism of each baseline.

8. In lines 8-10 of Algorithm 3, how to choose which bucket to append the data?

9. The idea of using model merging to expand experts is interesting, but the performance improvement is marginal according to Table 2.

---
Update After the Rebuttal Discussion Period:

The author has addressed most of my concerns, so I increased the scores accordingly.

**Questions:**

Please refer to Weakness.

---

> ### Author Response · Authors · 2024-11-20
>
> Thanks for the valuable comments that could help us improve the work. Next, we will respond to the questions, and we would appreciate it if you could raise the scores when the concerns are addressed.
>
> > **W1: It is unclear how many samples each method uses during training in Table 1.**
>
> In the 'Dataset' subsection of 'EXPERIMENTS' (Lines 318-321), we state that UpIT reuses the dataset utilized in the previous baseline method [1], which contains a total of 500k samples. In subsection 3.3 'SCALING THE TRAINING DATASET' and Figure 3, we also validate UpIT and vanilla upcycling approaches by randomly sampling 50K, 100K, and 200K samples from the full 500K dataset, to assess the data-efficient nature of UpIT.
>
> > **W2: What would be the results of not using intermediate checkpoints to initialize MoE?**
>
> Not using intermediate checkpoints is the vanilla upcycling method (as stated in Lines 044-048), which often requires a lot of data during the training phase and the results are not satisfactory.
>
> In the 'Baselines' subsection of 'EXPERIMENTS' (Lines 312-316), we introduce two baselines under LoRA-based setting: $LoRAMoE_{PT}$ and $LoRAMoE_{SFT}$, as well as two baselines under FFN-based setting: $Upcycle_{PT}$ and $Upcycle_{SFT}$. A thorough explanation of these baselines is provided in Appendix A.1 (Lines 765-767, 769-771). These four baselines represent two approaches: one that employs the pre-trained model directly for upcycling, and another that conducts SFT initially and subsequently utilizes the final checkpoint for upcycling. Table 2, Figure 3, and Figure 4 show that UpIT significantly outperforms existing methods, whether in scenarios with sufficient or insufficient data, exhibiting outstanding flexibility, scalability, and performance upper bound.
>
> > **W3: What is the most important component that contributes to the final performance?**
>
> We first reiterate that the utilization of intermediate checkpoints and the process of route initialization complement each other and are both core contributions in UpIT (as stated in Contribution, Lines 131-141).
>
> In this paper, we aim to emphasize the importance of expert discrepancy in upcycling and incorporate the idea into the entire design of UpIT. Figure 2 highlights different checkpoints excel at different benchmarks, which introduces our first contribution of using intermediate checkpoints for expert preparation and corresponding expert expansion strategies. On this basis, we propose a router initialization stage with expert-specific data selection, utilizing and amplifying the differences among experts in the initialization phase. Overall, the initial differences among experts are the foundation of UpIT, and we fully utilize and amplify these differences to improve performance and data efficiency.
>
> > **W4: None of the baselines belong to specialized upcycling.**
>
> As described in Lines 48-51, specialized upcycling first trains a specific expert for each domain, which lacks flexibility as it is difficult to divide a mixed general dataset into separate domains. Even if we manage to split the dataset, the number of experts will be the same as the number of domains, making it hard to apply in practical scenarios as well, because MoE typically involves 8 or more experts, it is difficult to guarantee that every training data has so many domains.
>
> In constant, UpIT considers more universal scenarios, in which data comes from a mixed set and cannot be effectively divided. Traditional specialized upcycling methods are no longer applicable, while UpIT achieves efficient and flexible upcycling by seeking natural differentiation between intermediate checkpoints.
>
> > **W5: In line 258, "merging all the expert models" seems more like a concatenation.**
>
> Thanks for pointing this out. Indeed, we meant "concatenation" here, and we will correct this in the next version.
>
> > **W6: The contents in Figure 1 are very small.**
>
> Thanks for pointing this out. We will correct this in the next version.
>
> > **W7: In Sec 3.1, it would be great to explain the mechanism of each baseline.**
>
> Thanks for pointing this out. Due to space limitations, we provide a detailed explanation and clarification in Appendix A.1 (DETAILED DESCRIPTION OF BASELINES, Lines 758-771), which are also mentioned in the the 'Baselines' subsection of 'EXPERIMENTS' in the main text (Line 316).
>
> [1] Wu et al. Parameter-Efficient Sparsity Crafting from Dense to Mixture-of-Experts for Instruction Tuning on General Tasks. EMNLP. 2024.

---

> ### Author Response · Authors · 2024-11-20
>
> > **W8: In lines 8-10 of Algorithm 3, how to choose which bucket to append the data?**
>
> As stated in Line 5 of Algorithm 3, we use perplexity (PPL) to measure the competence of each expert for each sample. Specifically, for each data, we determine the expert model with the lowest PPL and allocate the data to the bucket of that expert model. If the bucket is full, we then allocate the data to the bucket of the expert model with the second-lowest PPL, then, continue this process in order until the data is assigned.
>
> > **W9: The idea of using model merging to expand experts is interesting, but the performance improvement is marginal.**
>
> The core objective of model merging is to provide a method for achieving an instruction model with a flexible number of experts when intermediate checkpoints (i.e., experts) are scarce. The performance with expert expansion is very close to that of using more checkpoints, which means that we can achieve similar results with fewer experts. It is quite ideal result and fully demonstrates the effectiveness of expert extension (as mentioned in Lines 481-485).
>
> ---
>
> Thank you once again for your valuable feedback we are committed to refining our work based on your insights.
>
> If there are any remaining concerns or questions that we have not yet addressed, please do not hesitate to let us know. We are more than willing to provide any additional materials or clarifications that may be necessary.

---

> > ### Comment · Reviewer_DUkz · 2024-11-24
> >
> > Thank the authors for the reply.
> >
> > Regarding W4, it is still strange that none of these baselines are reported. I thought people could train a router over a training set to learn how to use the experts, which they developed in the first step of UpIT. Just like [1] did, this approach also seems to have a certain generalizability.
> >
> > [1] Muqeeth, M., Liu, H., Liu, Y., & Raffel, C. (2024). Learning to Route Among Specialized Experts for Zero-Shot Generalization. ArXiv, abs/2402.05859.

---

> > > ### Author Response · Authors · 2024-11-25
> > >
> > > Thank you for your valuable comments. In W4, you mentioned that "Also, UpIT is highly related to 'Specialized Upcycling' as the training requires curated expert data." We would like to clarify this point and highlight the motivation of our paper.
> > >
> > > The existing specialized upcycling methods **require a large amount of carefully curated expert data to construct specialized experts.** However, the UpIT method in this paper **does not need such carefully curated expert data.**
> > >
> > > In lines 318 - 322, within the 'Dataset' paragraph, we adopted the same data as that in [1]. This data is **a mixture of various domain data, which is the most common form of data organization in post-training.** Unlike traditional specialized Upcycling, we **do not need to separate different domain data and train them separately.**
> > >
> > > The primary contribution of UpIT is to construct discrepant experts using intermediate checkpoints, and then combine the PPL-based data selection strategy and the router initialization stage to obtain specialized experts, thereby achieving the effect of specialized upcycling with **non-carefully curated mixed data.**
> > >
> > > [1] Wu et al. Parameter-Efficient Sparsity Crafting from Dense to Mixture-of-Experts for Instruction Tuning on General Tasks. EMNLP. 2024.

---

> > > > ### Author Response · Authors · 2024-11-29
> > > >
> > > > We want to highlight once again that UpIT does not require meticulously constructed specialized data, nor does it need to use different domain data to train domain-specific experts separately.
> > > >
> > > > However, to address your concerns, we have trained a specialized MoE model following the method in [1].
> > > >
> > > > Specifically, we divide our data into four parts, including 100K from MetaMathQA, 100K from Magicoder, and two parts of 150K from SlimORCA, to train four domain-specific experts separately, followed by upcycling and post-training. The results are shown below:
> > > >
> > > > | Method | HumanE. | GSM8K   | HellaS. | BBH    | MMLU   | NQ     | Tri.QA | ARC_c  | ARC_e  | Avg |
> > > > |--------|---------|---------|---------|--------|--------|--------|--------|---------|---------|---------|
> > > > | Specialized (4E,A2) | **34.82** | 46.93 | 63.81 | **39.06** | 48.37 | 23.94 | **47.02** | 62.88 | **83.05** | **49.99** |
> > > > | UpIT (4E,A2) | 34.21 | **47.20** | **63.92** | 38.79 | **49.15** | **24.07** | 46.38 | **63.72** | 81.93 | 49.93 |
> > > >
> > > > It can be observed that UpIT does not have a significant performance gap compared to specialized upcycling, but specialized upcycling relies on meticulously constructed domain-specific data and a more complex training process, which limits its application. In contrast, UpIT can directly learn from mixed data and reinforce discrepancies among experts.
> > > >
> > > > ---
> > > >
> > > > If we have addressed your concerns, we would be delighted if you could consider raising the score.

---

> > > > > ### Author Response · Authors · 2024-12-03
> > > > >
> > > > > Dear reviewer,
> > > > >
> > > > > As the discussion period is nearing its conclusion, we earnestly hope to receive your feedback on our paper. We are keenly aware of the valuable insights you can provide, and we are hopeful that any outstanding concerns or questions will be communicated to us promptly. Your comments are crucial for the finalization of our work, and we are prepared to offer any further materials, experimental details, or clarifications that you may require.
> > > > >
> > > > > We are deeply appreciative of your expertise and are confident that your feedback will significantly contribute to the enhancement of our research. We look forward to your response before the discussion period ends.
> > > > >
> > > > > Best regards,
> > > > >
> > > > > The authors,

---

### Official Review · Reviewer_bbSt · 2024-11-05

**Soundness:** 2
**Presentation:** 3
**Contribution:** 3
**Rating:** 6
**Confidence:** 4

**Summary:**

The authors propose the Upcycling Instruction Tuning (UpIT) method to transform a pre-trained dense model into an MoE instruction tuned model. A primary concern in the standard upcycling approach is how to promote diversity among the experts at the initialization stage to improve results. UpIT achieves this by saving a series of intermediate checkpoints at fixed intervals during the fine-tuning of the dense pre-trained model. These checkpoints are used as initializations for the expert weights, followed by the curation of expert-specific data subsets to pre-optimize routing weights. The final step involves merging all expert models and routing vectors to form the MoE model, which is then fine-tuned on the entire instruction dataset.

The proposed method is implemented for Llama 2 7B and Sheared Llama 2.7B. The authors present the results for different amount of training data, number of Experts, LoRA vs full model training. They also conduct ablation studies on the checkpoint selection, parameter merging, and data selection strategies for router initialization.

**Strengths:**

- The paper is generally well-written, providing a clear explanation of the UpIT method.
- Introducing MoEs at the fine-tuning stage with limited training data is significant and impactful.
- The ablation experiments offer valuable insights into the method's effectiveness.

**Weaknesses:**

- **Expert Diversity:** While the UpIT method enhances expert diversity at initialization, Figure 2's motivation could be more robust. The way the motivation is provided in this figure suggests that different checkpoints excel at different benchmarks. I am wondering to what extend this argument is correct. For example on Factual knowledge, checkpoints saved at 0.25 and 1.5 Epochs have merely identical performance for the MMLU and ARC-e tasks. It also appears that the performance of the model does not improve much across these benchmarks over the fine-tuning stage to indicate the model is focusing on some tasks more than others over the course of training. A stronger case could be made if task interference led to performance drops of some tasks over other tasks during training, suggesting earlier checkpoints might be better for initializing experts whose associated tasks are dominated by other tasks over time.

- **Checkpointing Mechanism:** Given its centrality to UpIT, distinguishing the impact of initialization strategy from routing initialization and data selection is crucial. Table 2 suggests that much of the performance gain stems from router initialization. It would be beneficial to assess the effect of the proposed router initialization on other upcycling methods, such as naive upcycling via cloning with noise.

- **Expert Expansion:** The ablation study in Table 2 indicates that the DARE-based expert expansion algorithm does not significantly outperform simply saving more checkpoints. This aspect may not be a substantial contribution to the work.

**Questions:**

**Routing Distributions:** I find the routing distributions to different experts shown in Fig 5 hard to explain. To my understanding, there is no explicit mechanism during the initialization and MoE training stages that would encourage such an extreme specialization effect. The checkpoints saved at different steps are not specifically task conditional and in practice a vast majority of the samples coming from different benchmarks would be fine across different checkpoints. It is also unlikely that the acquired subsets in $D_s$ are purely task-specific samples. Can the authors provide insight into why this pattern appears?

---

> ### Author Response · Authors · 2024-11-20
>
> We are encouraged that you find UpIT is **significant and impactful**. Thanks for the valuable comments that could help us improve the work. We will respond to the questions, and *we would appreciate it if you could kindly raise the scores when the concerns are addressed.*
>
> > **W1 (Expert Diversity): While the UpIT method enhances expert diversity at initialization, Figure 2's motivation could be more robust.**
>
> In this paper, we aim to emphasize the importance of expert discrepancy in upcycling and incorporate the idea into the entire design of UpIT. Figure 2 first highlights the different checkpoints during instruction tuning that are inherently suitable for constructing specialized experts, since different checkpoints excel at different benchmarks, which introduces our first contribution of using intermediate checkpoints for expert preparation and corresponding expert expansion strategies. On this basis, we propose a router initialization stage with expert-specific data selection, utilizing and amplifying the differences among experts in the initialization phase. Overall, the initial differences among experts are the foundation of UpIT, and we fully utilize and amplify these differences to improve performance and data efficiency.
>
> Note that we present the results of multiple benchmarks in a single diagram, with a wide y-axis range, making it visually appear that the effects of different checkpoints are similar. However, in reality, they have significant differences (Table 6,7 provide more accurate numerical comparisons during an instruction tuning stage).
>
> > **W2 (Checkpointing Mechanism): The impact of the initialization strategy from routing initialization and data selection is crucial.**
>
> We first reiterate that the utilization of intermediate checkpoints and the process of route initialization complement each other and are both core contributions in UpIT (as stated in Contribution, Lines 131-141). In vanilla upcycling, all experts are the same in the initial stage, which does not apply to expert-specific data selection and therefore cannot perform the routing initialization operation.
>
> The table below extracts from Table 1 and Table 2 (right) demonstrate that roughly upcycle discrepant experts without proper router initialization do not yield good performance, because a randomly initialized router lacks the awareness of the experts, and direct training inadvertently reduces the discrepancy between them. Considering that noise usually follows a Gaussian distribution, introducing noise to the same is similar to the UpIT (Random) setting, which also cannot achieve satisfactory results.
>
> | Method               | Average |
> |----------------------|---------|
> | LoRAMoE_{SFT}    | 49.99 |
> | UpIT (w/o Init)      | 49.96 |
> | UpIT (Random)     | 49.30 |
> | UpIT (Skilled)       | 52.21 |
>
> > **W3 (Expert Expansion): Expert expansion does not significantly outperform simply saving more checkpoints.**
>
> The core objective of model merging is to provide a method for achieving an instruction model with a flexible number of experts when intermediate checkpoints (i.e., experts) are scarce. The performance with expert expansion is very close to that of using more checkpoints, which means that we can achieve similar results with fewer experts. It is an quite ideal result and fully demonstrates the effectiveness of expert extension. By the way, DARE is not the focus of this paper, alternative parameter merging methods, such as average merging, could also be employed.
>
> > **Q1 (Routing Distributions): Why does such a highly skewed router distribution arise?**
>
> All processes of UpIT emphasize the discrepancy of experts. Building on the discrepancy between experts, we propose an expert-specific data selection to filter the most skilled data for each expert. In the router initialization phase, we pre-define the allocation strategy of the router to achieve expert-aware routing, which will further enhance the differences between experts in subsequent post-training. In this way, the final router distribution exhibits strong preferences.
>
> ---
>
> Thank you once again for your valuable feedback we are committed to refining our work based on your insights.
>
> If there are any remaining concerns or questions that we have not yet addressed, please do not hesitate to let us know. We are more than willing to provide any additional materials or clarifications that may be necessary.

---

> > ### Author Response · Authors · 2024-11-29
> >
> > Dear reviewer,
> >
> > Thank you very much for the time and effort you have dedicated to reviewing our work.
> >
> > If our response satisfies you, we anticipate your reconsideration of the overall assessment of our paper. If there are any remaining concerns or questions that we have not yet addressed, please do not hesitate to let us know. We are more than willing to provide any additional materials, experimental details, or clarifications that may be necessary.
> >
> > We greatly value your insights and believe they will substantially enhance the quality of our work.
> >
> > Best regards,
> >
> > The authors,

---

> > > ### Author Response · Authors · 2024-12-03
> > >
> > > Dear reviewer,
> > >
> > > As the discussion period is nearing its conclusion, we earnestly hope to receive your feedback on our paper. We are keenly aware of the valuable insights you can provide, and we are hopeful that any outstanding concerns or questions will be communicated to us promptly. Your comments are crucial for the finalization of our work, and we are prepared to offer any further materials, experimental details, or clarifications that you may require.
> > >
> > > We are deeply appreciative of your expertise and are confident that your feedback will significantly contribute to the enhancement of our research. We look forward to your response before the discussion period ends.
> > >
> > > Best regards,
> > >
> > > The authors,

---

### Author Response · Authors · 2024-11-20

### General Response

We sincerely thank all the reviewers for their efforts and valuable feedback. Many comments are essential to improving the quality of our paper. We are encouraged that the reviewers find UpIT an interesting and solid work, a very promising and innovative approach and that the experiments are extensive and well-explained.

Considering that **"vanilla upcycling requires large amounts of data for training"**, and **"specialized upcycling is data-efficient but lacks flexibility in scaling the number of domain-specific experts"**, UpIT aims to answer the question of **"how to efficiently and flexibly train a MoE instruction model based on a dense pre-trained model?"**

Our pilot experiment (Figure 2) reveals that the intermediate checkpoints saved during instruction tuning naturally exhibit discrepancies, with different checkpoints excelling at different tasks. To address the need for more discrepant experts when checkpoints are insufficient, we propose an expert expansion stage with a genetic algorithm, which merges new experts from existing ones to achieve flexible upcycling. Leveraging these discrepancies between experts, we propose an expert-specific data selection algorithm that allows checkpoints (i.e., experts) to select the data they perform best on. Next, through an innovative router initialization stage, we enhance the discrepancy between experts and pre-define the router allocation strategy. In scenarios with sufficient or insufficient data, UpIT exhibits outstanding flexibility, scalability, and performance upper bound.

**In one word, This paper aims to emphasize the importance of expert discrepancy in upcycling and incorporate the idea into the entire design of UpIT.**

We highlight the motivation and contributions of UpIT here and sincerely hope that the reviewers can gain a clearer, more accurate, and deeper understanding of our work.

---

### Meta-Review · Area_Chair_UF4o · 2024-12-13

**Metareview:**

After careful reading, I recommend its rejection due to fundamental concerns about the clarity, contribution significance, and soundness of the proposed methodology.

Firstly, while the paper introduces the novel idea of leveraging intermediate checkpoints to initialize experts for Mixture-of-Experts (MoE) models, its motivation and execution lack robustness. The claim that intermediate checkpoints inherently lead to meaningful specialization remains largely unsupported by the evidence provided. Figures and tables, such as Figure 2, demonstrate only marginal differences in performance across checkpoints for certain benchmarks, contradicting the authors’ assertion that these checkpoints exhibit task-specific discrepancies. Moreover, the reliance on heuristic-based checkpoint selection, without a principled explanation or empirical demonstration of its generalization across datasets, weakens the paper’s broader applicability and reproducibility. This is especially problematic in scenarios with highly varied or out-of-distribution datasets, which are common in real-world applications.

Secondly, the proposed expert expansion strategy, which combines genetic algorithms and parameter merging, adds complexity without substantial performance gains. The ablation studies suggest that simply saving more checkpoints performs comparably, raising doubts about the necessity of the proposed approach. The lack of clarity regarding the trade-offs between diversity and scalability in expert merging (parameters m and n) further limits the practical guidance offered to practitioners. Additionally, the router initialization mechanism, while innovative, is insufficiently compared to simpler or alternative upcycling methods. This leaves unanswered questions about its relative effectiveness and undermines the claimed contributions.

**Additional Comments On Reviewer Discussion:**

During the rebuttal period, reviewers raised several concerns: (1) the marginal performance differences across checkpoints undermining the claim of inherent task-specific specialization, (2) the lack of clarity on how checkpoint intervals and diversity/scalability trade-offs affect performance, (3) the limited improvement from the expert expansion strategy compared to simpler baselines, and (4) insufficient generalization evidence across datasets.
The authors provided additional analyses and experiments, showing incremental performance gains with more experts and addressing some points on checkpoint selection strategies. However, they failed to fully substantiate their core claims of expert specialization or the necessity of their methods, with explanations often relying on heuristics rather than rigorous evidence. While the rebuttal addressed presentation issues and added experimental details, the key methodological and generalizability concerns persisted.

---

### Decision · Program_Chairs · 2025-01-22

Reject